# Modeling Stochastic Conditional Dynamics from Sparse Observations via Kernel-Stabilized Flow Matching

**Adam P. Generale**  *agenerale3@gatech.edu*
*Georgia Institute of Technology*

**Andreas E. Robertson**  *aerober@sandia.gov*
*Sandia National Laboratories*

**Surya R. Kalidindi**  *surya.kalidindi@me.gatech.edu*
*Georgia Institute of Technology*

**Reviewed on OpenReview:** *https://openreview.net/forum?id=3A6oAS2TWo*

## Abstract

Learning to transform conditional probability densities over time is a fundamental challenge spanning probabilistic modeling and the natural sciences. This task is paramount when forecasting the evolution of stochastic nonlinear dynamical systems in biological and physical domains. While flow-based models can predict the temporal evolution of probability distributions, existing approaches often assume discrete conditioning with samples that are paired across time, limiting their scientific applicability where frequently only sparse data with unpaired continuous conditioning is available. We propose *Conditional Variable Flow Matching* (CVFM), a framework for learning flows transforming conditional distributions with amortization across the continuous space of conditional densities. CVFM addresses the high-variance instability of prior methods by jointly sampling flows over state and conditioning variables, utilizing a conditioning mismatch kernel alongside a conditional Wasserstein distance to reweight the conditional optimal transport objective. Collectively, these advances allow for learning dynamics from sparse unpaired measurements of state-condition across time. We evaluate CVFM on conditional mapping benchmarks and a case study modeling the temporal evolution of materials internal structure during manufacturing processes, observing improved performance and convergence characteristics over existing conditional variants. Code is available at `https://github.com/agenerale/conditional-variable-flow-matching`.

## 1 Introduction

Appropriately modeling the time-dependent evolution of distributions is a central goal in multiple scientific fields, such as single-cell genomics (Tong et al., 2023a; Bunne et al., 2023a), meteorology (Fisher et al., 2009), robotics (Ruiz-Balet & Zuazua, 2023; Chen et al., 2021), and materials science (Kalidindi, 2015; Adams et al., 2013). Across the sciences, forecasting stochastic nonlinear dynamical systems requires a methodology for learning the transformations of time-evolving densities. For many applications, this capability must be achievable given only *unpaired* observational samples[1], or observations across time which are not in correspondence. This requirement arises due to practical constraints on data collection in such applications. For example, in both single-cell genomics and materials science, experimental testing to quantify the system's state is often destructive, precluding measurement of the state of a single sample across multiple time steps (Tong et al., 2023a; Bunne et al., 2023a; Ghanavati & Naffakh-Moosavy, 2021; ASTM International, 2024).

---

[1]For unconditioned distributions, *unpaired* samples refers to the ability to sample from time marginal distributions independently, $x_0 \sim p_0(x)$ and $x_1 \sim p_1(x)$, rather than the joint, $p(x_0, x_1)$.

Various approaches to address this challenge have recently been proposed, including diffusion Schrödinger bridges (DSB) (Liu et al., 2022a; Chen et al., 2023; De Bortoli et al., 2021; Bunne et al., 2023a; Tang et al., 2024) alongside extensions of Flow Matching (FM) (Tong et al., 2023b;a). These approaches generalize denoising diffusion probabilistic models (Ho et al., 2020), score matching (Song et al., 2021), and FM (Lipman et al., 2023; Albergo & Vanden-Eijnden, 2023; Liu et al., 2022b), to arbitrary source distributions – a necessary relaxation to model the evolutionary pathways of physical or biological systems, as such natural systems rarely exhibit Gaussian source distributions[2].

Despite the apparent success of such approaches, their practical utility has been limited, as they solely permit the simulation of evolving unconditional distributions. However, in modeling the dynamics of real systems, the most critical questions often take the form of *how might an intervention affect the resulting dynamics*? For example, the dynamics of materials manufacturing processes depend upon process parameters such as applied temperature or power (Liu et al., 2022c; Schrader & Elshennawy, 2000). Such questions require modeling the evolution of conditional distributions. In addition to unpaired state measurements, we often face the challenge of unpaired conditioning, which we define as the scenario in which data observed at discrete time frames possesses conditioning variables which are similarly unpaired. Formally, observed samples are drawn independently from their respective joint distributions at the initial and final time steps[3] as $(x_0, y_0) \sim p_0(x, y), (x_1, y_1) \sim p_1(x, y)$ where there is no guarantee that $y_0 = y_1$, distinguishing this from traditional conditional settings where a fixed $y$ is observed. Aside from the destructive nature of common data collection in the sciences, unpaired conditioning commonly arises due to the prohibitive costs of sample acquisition. Design-of-experiments or active learning approaches are frequently employed to mitigate these costs by minimizing the number of experiments – identifying a series of maximally informative tests (Lookman et al., 2019; Tran et al., 2020). This diversity ensures that conditioning is purposefully rarely repeated.

Extensions modeling the dynamics of conditional distributions are rapidly developing (Ho & Salimans, 2022; Zheng et al., 2023; Bunne et al., 2022; Harsanyi et al., 2024; Bunne et al., 2023b; Isobe et al., 2024). Conditional input convex neural networks (ICNN) (Bunne et al., 2022; Harsanyi et al., 2024; Bunne et al., 2023b) and initial conditional extensions of flow matching (Isobe et al., 2024; Zheng et al., 2023; Dao et al., 2023), in particular, require datasets with matching conditioning, or the ability to select $y \in \mathcal{Y}$ and subsequently sample $x_t \sim p_t(x|y)$. This structure degrades in the limit of continuous conditioning variables, where obtaining multimarginal samples with equivalent conditioning is infeasible. While select recent works have begun to explore extensions enabling such continuous conditional treatment through conditional optimal transport (OT) between densities, these formulations suffer from high-variance instabilities (Chemseddine et al., 2024; Kerrigan et al., 2024; Baptista et al., 2024; Tabak et al., 2019). We demonstrate that in sparse data regimes – common in scientific discovery – standard mini-batch approximations of the conditional coupling lead to variance explosion, rendering training unstable or non-convergent. To address this, we introduce a conditioning mismatch kernel which explicitly enforces local consistency on the conditioning manifold. This addition coupled with the conditional Wasserstein distance combines to significantly increase the algorithm's stability and performance.

**We propose *Conditional Variable Flow Matching* (CVFM), a general approach for learning the flow between source and target conditional distributions**. Importantly, CVFM supports *entirely unpaired datasets*, wherein neither the sample data nor their corresponding conditioning variables need to be paired. We motivate the proposed CVFM algorithm through a theoretical analysis of the stability of flow matching on conditional densities, formally establishing the conditional identity coupling as a prerequisite for consistent vector field learning. To address this, core to CVFM is the usage of two conditional flows, a conditional Wasserstein distance, and a condition-dependent mismatch kernel, generalizing and stabilizing existing simulation-free objectives for continuous normalizing flows (CNF) (Lipman et al., 2023; Albergo & Vanden-Eijnden, 2023) and stochastic dynamics (Tong et al., 2023a; Shi et al., 2023) to the conditional setting. We specifically focus on dynamics wherein the marginal distribution over the conditioning variable remains constant – a common setting in applied problems. The algorithm is introduced in the dynamical

---

[2]We note that when approximating the dynamics of real systems, the model is trained to transform the state density between successive time steps. Therefore, even if the density at $t = 0$ is Gaussian, the bridge between arbitrary time steps must be generalized.

[3]These times can be fictitious (as in standard generative models, (Tong et al., 2023b)) or can refer to the real time between time steps (as in dynamics, (Tong et al., 2023a)).

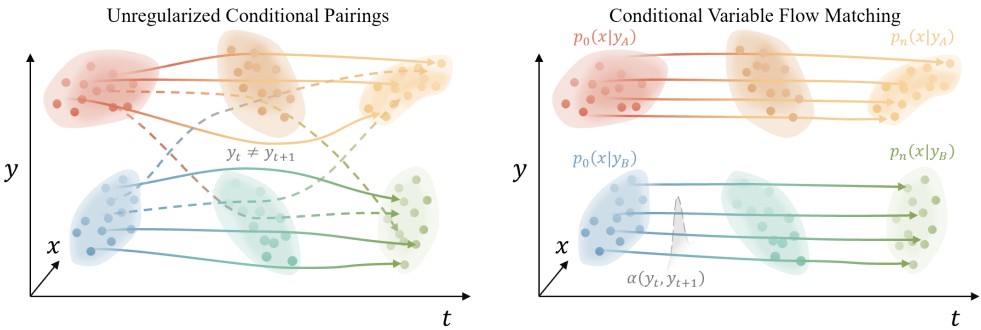

Figure 1. **The stability gap in learning conditional dynamics from unpaired data. (Left) Unregularized Conditional Pairings:** Standard conditional flow matching methods employ couplings (such as naïve or conditional OT $\pi_\eta((x,y),(x',y'))$) that **generate pairings without enforcing local conditional consistency**. In sparse data regimes, these unregularized couplings often propose pairings between samples with disparate conditioning values (e.g., coupling $p(x|y_A)$ to $p(x|y_B)$ where $y_A, y_B$ denote conditioning variables drawn from disjoint subsets $\mathcal{Y}_A, \mathcal{Y}_B \subset \mathcal{Y}$). This results in high-variance, mismatched flows where the dynamics erroneously jump across the conditioning manifold ($y_t \neq y_{t+1}$). **(Right) Conditional Variable Flow Matching (CVFM):** Our framework integrates a conditioning mismatch kernel $\alpha(y_t, y_{t+1})$ with the conditional Wasserstein distance in the training objective. The kernel acts as a soft filter, suppressing the mismatched trajectories proposed by the OT solver while retaining locally consistent pairings. The result is a conditionally stable, laminar vector field that respects the underlying conditional distributions.

formulation of optimal transport (OT), augmenting FM and ordinary differential equation (ODE) based transport in defining a straightforward training objective for learning amortized conditional vector fields. Our objective facilitates simulation across the conditional density manifold, leveraging these mechanisms to enable robust conditional OT. Subsequently, we analyze CVFM on several toy problems, demonstrating its improved performance and convergence behavior compared to existing methods. Beyond synthetic benchmarks, we showcase the scientific utility of CVFM by modeling the complex stochastic evolution of material microstructures. By successfully forecasting phase-field simulated dynamics conditioned across a continuous space of manufacturing parameters, we demonstrate the framework's capability to capture realistic physical phenomena. Across these evaluations, we further demonstrate the applicability of our method to approximating conditional Schrödinger bridges with a score-based stochastic differential equation (SDE) extension to FM (Tong et al., 2023a).

## 2 Dynamic Mass Transport Methods

### 2.1 Flow Matching

Continuous Normalizing Flows (CNF) (Chen et al., 2018) define a mapping between distributions $x_0 \sim p_0(x)$ and $x_1 \sim p_1(x)$ on the same domain, $x_0, x_1 \in \mathbb{R}^N$ via the following ordinary differential equation (ODE).

$$\frac{d}{dt}\phi_t(x) = u_t(\phi_t(x)), \quad \phi_0(x) = x \tag{1}$$

Individual samples $x_0 \sim p_0(x)$ can be transformed to $x_1 \sim p_1(x)$ by integrating the vector field $u_t : [0,1] \times \mathbb{R}^N \to \mathbb{R}^N$ and solving the ODE in Eq. (1).

Flow matching (FM) provides a simulation-free objective for constructing the *marginal probability path* $p_t(x)$ via a marginalization of sample conditioned probability paths $p_t(x|z)$, conditioned on observations $z = (x_0, x_1)$ drawn from the empirical distributions – distributions defined implicitly by a set of observed data at the initial and final time steps, $q(x_0)$ and $q(x_1)$. Lipman et al. (2023) demonstrates that one can similarly marginalize over conditional vector fields $u_t(x|z)$ to construct $u_t(x)$, generating the probability flow

$p_t(x)$ (Theorem 1 (Lipman et al., 2023)). The consequences of which permit directly regressing upon the conditional vector field as

$$\mathcal{L}_{\text{CFM}}(\theta) = \mathbb{E}_{t,q(z),p_t(x|z)} \|v_\theta(x,t) - u_t(x|z)\|^2 \tag{2}$$

where for conditional Gaussian paths $p_t(x|z) = \mathcal{N}(x|\mu_t(z), \sigma_t^2(z))$ of $\phi_{t,z}(x) = \mu_t(z) + \sigma_t(z)x$ the unique conditional vector field $u_t(x|z)$ can be solved in closed form (Theorem 3 (Lipman et al., 2023), Theorem 2.1, (Albergo & Vanden-Eijnden, 2023)).

## 2.2 Optimal Transport

The optimal transport (OT) problem aims to identify a mapping between measures, $\nu$ and $\mu$, with minimal displacement cost (Villani, 2009). The Kantorovich relaxation attempts to recover the OT coupling $\pi$ given the potential set of all couplings on $\mathcal{X} \times \mathcal{Y}$, such that the coupling's marginals are the original distributions, $\Pi(\mu, \nu) = \{\pi \in \mathcal{P}(\mathcal{X} \times \mathcal{Y}) : P_{\mathcal{X}\#}\pi = \mu \text{ and } P_{\mathcal{Y}\#}\pi = \nu\}$. The resulting distance $W(\mu, \nu)$ is the Wasserstein distance

$$W(\mu, \nu) = \inf_{\pi \in \Pi(\mu,\nu)} \int c(x,y)d\pi(x,y) \tag{3}$$

where the squared Wasserstein-2 distance $W(\mu, \nu)_2^2$ is induced by the ground cost $c(x,y)^2$. Eq. (3) can be made computationally efficient by adding a KL penalty between the coupling $\pi$ and the independent joint $\mu \otimes \nu$, leading to the Sinkhorn formulation, or entropically-regularized OT (Cuturi, 2013).

**OT-FM**: While the Gaussian conditional probability paths for $p_t(x|z)$ in FM are the OT paths after conditioning on $z$ (Peyré & Cuturi, 2020), the induced marginal flow, defining $p_t(x)$, does not provide OT between distributions. Recent works have demonstrated that approximate *dynamic* marginal OT can be achieved through identifying the *static* OT map within mini-batches, and resampling $q(z) = \pi^*(x_0, x_1)$ according to the OT coupling $\pi^*$(Tong et al., 2023b; Pooladian et al., 2023).

## 2.3 Schrödinger Bridge

The Schrödinger bridge problem aims to identify the most likely stochastic mapping between arbitrary marginal distributions $\mathbb{P}_0 = \mu_0$ and $\mathbb{P}_1 = \mu_1$ with respect to a given reference process $\mathbb{Q}$ (Cuturi, 2013; Léonard, 2013; Schrödinger, 1932), defined as $\mathbb{P}_t^* = \text{argmin}_{\mathbb{P}_0=\mu_0,\mathbb{P}_1=\mu_1} \text{KL}(\mathbb{P}_t\|\mathbb{Q}_t)$. Frequently, $\mathbb{Q}$ is taken to be $\mathbb{Q} = \sigma\mathbb{W}$, where $\mathbb{W}$ is standard Brownian motion, otherwise known as the *diffusion Schrödinger bridge* (De Bortoli et al., 2021; Vargas, 2021; Bunne et al., 2023a; Shi et al., 2023).

Prior work has elucidated a rich relationship between the Schrödinger bridge problem and OT, in particular entropy-regularized OT (Léonard, 2013; Mikami & Thieullen, 2006; Mikami, 2004; Léonard, 2010). More specifically, with the reference process assumed as standard Brownian motion, the marginals of the dynamic Schrödinger bridge can be considered to be a mixture of Brownian bridges weighted by the *static* entropic OT coupling, a construction reminiscent to the marginalization of probability paths in FM $p_t(x) = \int p_t(x|z)d\pi_\varepsilon^*(z)$ (Léonard, 2010; 2013).

## 2.4 Score and Flow Matching

Given the notable connection between the Schrödinger bridge problem (Schrödinger, 1932) and entropy regularized OT (Léonard, 2013; Mikami & Thieullen, 2006; Mikami, 2004; Léonard, 2010), a natural step to address this challenge in a simulation-free manner is to seek an extension to the OT-FM objective. Tong et al. (2023a) demonstrated the feasibility of this approach through generalizing Eq. (2) to simultaneously regress on the conditional drift and score of an SDE. The proposed method replaces the flow ODE with the general SDE $dx = u_t(x)dt + g(t)dw_t$ where $u_t(x)$ is the SDE drift, $dw_t$ is standard Brownian motion, and $g(t)$ is the diffusion coefficient (Tong et al., 2023a). The SDE drift and a corresponding ODE vector field $\hat{u}_t(x)$ are intimately related by the *probability flow* ODE of the process: $\hat{u}_t(x) = u_t(x) - \frac{g(t)^2}{2}\nabla \log p_t(x)$.

Mirroring the previous flow-based models, the vector field, $u_t(x)$, and the score function, $s_t(x) = \nabla \log p_t(x)$, defining the SDE's drift are unknown. Score and Flow Matching ($[\text{SF}]^2\text{M}$) involves generalizing FM, Eq.

(2), to fit approximators of both objects

$$\mathcal{L}_{[\text{SF}]^2\text{M}}(\theta) = \mathcal{L}_{\text{CFM}}(\theta) + \mathbb{E}_{t,q(z),p(x|z)}\lambda(t)^2\|s_\theta(x,t) - \nabla \log p_t(x|z)\|^2 \tag{4}$$

where $\lambda(t)^2$ is a time-dependent normalization factor ensuring uniform weighting of the score across time (Tong et al., 2023a; Song et al., 2021; Ho et al., 2020; Karras et al., 2022). Further background can be found in Appendix B.

We emphasize that the conceptual similarity between the derivation of the flow matching objective and that of generalized score and flow matching signifies that any improvements in the first can be readily transferred to the second. As a result, in this work we theoretically restrict ourselves to purely flow-based models to simplify communication of novel developments, while case studies also include score and flow-based models.

## 3 Conditional Variable Flow Matching

### 3.1 Constructing Conditional Probability Paths and Vector Fields

We first note that the original marginalization motivating the flow matching objective can be extended to construct a probability flow across both $x$ and a conditioning variable $y$ as

$$p_t(x|y) = \iint p_t(x|y,z,w)q(z,w)dzdw \tag{5}$$

where $q(z,w)$ now denotes the empirical distribution over $z = (x_0, x_1)$ and $w = (y_0, y_1)$, where samples are drawn from $x_0, y_0 \sim q(x_0, y_0)$ and $x_1, y_1 \sim q(x_1, y_1)$. We further assume that the conditional joint probability path decomposes as $p_t(x,y|z,w) = p_t(x|z)p_t(y|w)$, resulting in two simultaneous conditional flows, one over the state variable $x$ given marginal observations $z$, and the other over the conditioning variable $y$ given the associated marginal sampled conditioning variables $w$. In other words, we assume that the joint distribution at intermediate times, $p_t(x,y)$, retains a dependency between $x$ and $y$ as defined by the joint discrete time empirical distributions, $q(z,w)$.

We can also extend this line of thought towards defining a marginal conditional vector field, through marginalizing over vector fields conditioned on observations $z$ and $w$ as

$$u_t(x|y) = \mathbb{E}_{q(z,w)}\frac{u_t(x|z)p_t(x|z)p_t(y|w)}{p_t(x,y)} \tag{6}$$

where $u_t(x|z) : [0,1] \times \mathbb{R}^N \to \mathbb{R}^N$ is a conditional vector field generating $p_t(x|z)$ from $p_0(x|z)$, without any explicit dependence upon the conditional distribution over $y$. Following Theorem 3 (Lipman et al., 2023) and Theorem 2.1, (Albergo & Vanden-Eijnden, 2023), and Theorem 3.1 (Tong et al., 2023a), we prescribe a form to both conditional vector fields such that they generate their respective conditional probability distributions[4]. This method of combining conditional vector fields can be shown to generate the marginal conditional vector field, $u_t(x|y)$, which is formalized in the following theorem.

**Theorem 3.1** *The marginal conditional vector field Eq. (6) generates the marginal conditional probability path Eq. (5) from $p_0(x|y)$ given samples of $q(z,w)$ if $q(y_0) = q(y_1)$ and the conditioning variable pairs $(y_0, y_1)$ drawn from $q(z,w)$ follow the identity coupling.*

Equality of the empirical distributions refers to equality in the underlying distribution, not the sample sets which define them implicitly. This result deviates from prior results in standard Flow Matching by Lipman et al. (2023) (Theorem 1) and Albergo et al. (2023) (Theorem 2.6) as well as recent extensions for conditional densities (Kerrigan et al., 2024); the theorem states that OT over the conditioning variable is necessary for learning amortized conditional vector fields. Our framework is designed specifically to implement this conclusion. Through this lens, we will see the practical implications on learned mappings and training dynamics when this result is violated. The full proofs of all theorems are provided in Appendix A.

---

[4]For Gaussian probability paths with deterministic dynamics: $u_t(x|z) = x_1 - x_0$ (Pooladian et al., 2023; Tong et al., 2023b; Albergo & Vanden-Eijnden, 2023) and stochastic dynamics: $\hat{u}_t(x|z) = ((1-2t)/t(1-t))(x - (tx_1 + (1-t)x_0)) + (x_1 - x_0)$, $\nabla_x \log p_t(x|z) = (tx_1 + (1-t)x_0 - x)/(\sigma^2 t(1-t))$ (Tong et al., 2023a).

## 3.2 Flow Matching for Conditional Distributions

Even given Eq. (5) and Eq. (6), their incorporation in an overall objective for training a neural network approximator to $u_t(x|y)$ is still limited by several intractable integrals. Instead, we can obtain an unbiased estimator of the marginal conditional vector field and resulting probability path provided only samples from known distributions and the ability to compute $u_t(x|z)$ through the proposed conditional variable flow matching objective.

$$\mathcal{L}_{\text{CVFM}}(\theta) = \mathbb{E}_{t, q(z,w), p_t(x|z) p_t(y|w)} \left[ \alpha(w) \| v_\theta(x, y, t) - u_t(x|z) \|^2 \right] \tag{7}$$

The conditioning mismatch kernel $\alpha(w)$ dynamically modulates the acceptance of conditioning pairs, further facilitating conditional OT. The following theorem formalizes the use of the tractable objective in Eq. (7) as a close approximation to the ideal marginal objective.

**Theorem 3.2** *Let $p_t(x|y) > 0$ for all $x \in \mathbb{R}^N$, $y \in \mathbb{R}^M$, $t \in [0,1]$ and $\mathcal{L}_{MCFM}(\theta) = \mathbb{E}_{t, p_t(x,y)} \| v_\theta(x, y, t) - u_t(x|y) \|_2^2$. Let $\alpha$ be a stationary, isotropic kernel depending only on $r = \|y_0 - y_1\|_2$, decaying to 0 as $r \to \infty$, and $\mathcal{C}^2$ in a neighborhood of 0 with $\alpha(0) = 1$ and $\alpha'(0) = 0$, and further assume the coupling $\pi(w)$ satisfies $\mathbb{E}_{\pi(w)}[r^2] < \infty$. Then there exists a constant $C$ such that for every $\theta$,*

$$\left\| \nabla_\theta \mathcal{L}_{\text{CVFM}}(\theta) - \nabla_\theta \mathcal{L}_{\text{MCFM}}(\theta) \right\| \le C \, \mathbb{E} \left[ \|y_0 - y_1\|_2^2 \right]. \tag{8}$$

*If $\pi(w)$ enforces $\mathbb{E}_{\pi(w)}[\|y_0 - y_1\|_2^2] = O(\varepsilon^2)$ then the gradient error is $O(\varepsilon^2)$.*

Beyond the gradient bound of Theorem 3.2, the learned conditional flow inherits desirable regularity properties from the architecture. When $v_\theta$ is locally Lipschitz in both $x$ and $y$, the induced pushforward $p_t(x|y)$ varies continuously in $y$ in the Wasserstein sense (Theorem 3.3, Appendix A.3), justifying amortized conditional inference with $v_\theta(x, y, t)$.

## 3.3 Stabilizing and Accelerating Training

Unlike previous flow matching frameworks, the empirical distribution, $q(z, w)$, cannot be sampled arbitrarily (Lipman et al., 2023; Tong et al., 2023a). Samples must be drawn such that $w = (y_0, y_1)$ follows the identity coupling $\pi(y_0, y_1)$, the theoretical reasons for which are described in Appendix A. We introduce two components to achieve this: an anisotropic transport cost, and a conditioning mismatch kernel.

**Anisotropic Optimal Transport for Conditional Distributions**: We consider two probability distributions, a source $\mu$ and a target $\nu$, in the space $\mathcal{P}(\mathcal{X} \times \mathcal{Y})$, where $\mathcal{X} \subseteq \mathbb{R}^N$ and $\mathcal{Y} \subseteq \mathbb{R}^M$. We aim to find an OT plan $\pi \in \Pi(\mu, \nu)$ that primarily transports mass along the $\mathcal{X}$ dimension while penalizing transport along the $\mathcal{Y}$ dimension.

A hard constraint prohibiting transport in $\mathcal{Y}$ would require that for any pair of coupled points $((x, y), (x', y'))$ in the support of $\pi$, we must have $y = y'$. This is often infeasible, especially in the empirical setting with continuous data. Instead, we introduce a penalty into the ground cost, an approach also explored in concurrent work (Kerrigan et al., 2024; Chemseddine et al., 2024; Baptista et al., 2024). We define the anisotropic cost function $c_\eta : (\mathcal{X} \times \mathcal{Y})^2 \to \mathbb{R}^+$ as:

$$c_\eta((x, y), (x', y')) = \|x - x'\|_p^p + \eta \|y - y'\|_p^p \tag{9}$$

where $\eta > 0$ is a hyperparameter controlling the penalty for transport in $\mathcal{Y}$, and $\|x\|_p$ is the $p$-norm. In this context, we refer to the transport cost induced by this anisotropic ground cost $c_\eta$ as the conditional Wasserstein distance.

**Proposition 3.1** *Let $\mu, \nu \in \mathcal{P}(\mathbb{R}^N \times \mathbb{R}^M)$ have finite $p$-th moments, and let $\mu_{\mathcal{Y}}$ and $\nu_{\mathcal{Y}}$ be their respective marginal distributions on $\mathbb{R}^M$. Let $\pi_\eta$ be an OT plan for the cost $c_\eta$ in Eq. (9). In the limit as the penalty parameter grows, the expected transport cost in the $\mathcal{Y}$ dimension converges to the $p$-Wasserstein distance between the marginals:*

$$\lim_{\eta \to \infty} \mathbb{E}_{((x,y),(x',y')) \sim \pi_\eta} [\|y - y'\|_p^p]^{1/p} = W_p(\mu_{\mathcal{Y}}, \nu_{\mathcal{Y}}) \tag{10}$$

*If the marginals are identical ($\mu_{\mathcal{Y}} = \nu_{\mathcal{Y}}$), then this limit is zero (Villani, 2009; Peyré & Cuturi, 2020).*

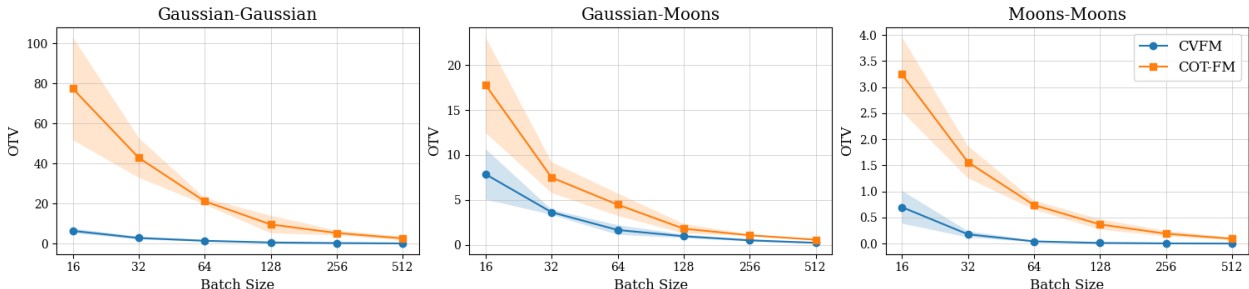

Figure 2. Impact of the kernel $\alpha(w)$ on estimator stability. Comparison of Objective Target Variance (OTV) $-\mathbb{E}_{t,q,p_t}||u_t(x|z)||^2$ – across batch sizes for the proposed CVFM and existing COT-FM frameworks (Kerrigan et al., 2024; Chemseddine et al., 2024). COT-FM exhibits explosive variance in sparse data regimes ($B = 16$). CVFM effectively suppresses this variance in all cases, demonstrating that the kernel is essential for stable training when dense sampling of the conditioning manifold is infeasible. Values were computed with $\eta = 10$ (the OT weight for the condition variable) over 5 seeds and are shown as $\mu \pm \sigma$.

In practice, we approximate the OT plan using mini-batches. In each iteration, we draw $k$ samples $(x_i, y_i)_{i=1}^k$ from the source $\mu$ and $k$ samples $(x'_j, y'_j)_{j=1}^k$ from the target $\nu$, forming empirical measures $\mu^k$ and $\nu^k$. The coupling is then estimated by solving the discrete OT problem between $\mu^k$ and $\nu^k$ with the cost $c_\eta$.

**Conditioning Mismatch Kernel**: While mini-batch OT approximates the correct coupling, it introduces significant estimator variance in regimes where the conditioning manifold is sparse. In such cases, samples with disparate conditioning values are paired, leading to noisy high-cost pairings which destabilize the vector field regression in Eq. (2). To mitigate this, CVFM incorporates a dynamic scaling term in the form of a stationary symmetric kernel $\alpha(w)$ into the objective Eq. (7). This term acts as a variance reduction mechanism by modulating the loss based on the degree of mismatch in the sampled conditioning variables $(y_0, y_1)$, effectively suppressing the contribution of mismatched pairings and promoting the conditional identity coupling, see Appendix E.2 for analysis. CVFM utilizes a squared exponential kernel, $\alpha(w) = \exp(-\|y_0 - y_1\|_2^2/2\sigma_y^2)$; however, our framework supports any stationary, positive-definite kernel that is a monotonically decreasing function of $\|y_0 - y_1\|_p$ (see Appendix E.2). As demonstrated in the following analysis, the kernel's presence stabilizes training – combating the need for long training schemes commonly required for flow matching methods – and improves overall performance. In particular, this addition is critical for ensuring model performance and training stability in sparse data regimes frequently encountered in scientific applications.

**Motivating Analysis**: To explicitly validate this, we analyze the Objective Target Variance (OTV), defined as $\mathbb{E}_{t,q,p_t}||u_t(x|z)||^2$, on a canonical 2D manifold with strong conditional dependence (see *8 Gaussians — 8 Gaussians* in Sec. 5). As illustrated in Figure 2, the anisotropic cost function Eq. (9) in isolation (COT-FM) exhibits variance explosion when batch sizes are small ($B = 16$), simulating the sparse data availability common in scientific applications. In this regime, the mini-batch OT solver is forced to couple samples with disparate conditioning values ($y \neq y'$) simply to satisfy marginal constraints, creating high-variance target velocities that contradict the local conditional geometry. By incorporating the kernel, CVFM suppresses this variance by an order of magnitude, ensuring stable convergence even when dense sampling of the conditioning variable is infeasible. Further analysis is provided in Appendix E. This empirical observation aligns with our theoretical insight; in COT-FM, the estimator variance scales with the volume of the conditioning space. The kernel effectively reduces the effective sample volume to a local neighborhood, bounding the variance.

To synthesize the theoretical components introduced above, we provide a unified training algorithm for CVFM in Algorithm 1. The framework easily adapts to both deterministic ODE paths and stochastic SDE Brownian bridges by modulating the target vector fields and adding a score-matching penalty, all while retaining the stabilizing benefits of the conditioning mismatch kernel $\alpha(w)$.

---

**Algorithm 1** Conditional Variable Flow Matching (CVFM & CVSFM)

---

**Require:** Source and target data distributions $q(x_0, y_0)$ and $q(x_1, y_1)$, noise hyperparameters $\sigma_x$, $\sigma_y$, mini-batch size $B$, drift network $v_\theta$, and (if SDE) score network $s_\theta$ and schedule $\lambda(t)$.

1: **while** Training **do**
2: $\quad \{(x_0, y_0), (x_1, y_1)\}_{b=1}^B \sim q(x_0, y_0), q(x_1, y_1)$
3: $\quad \pi \leftarrow \text{OT}(\{(x_0, y_0), (x_1, y_1)\}_{b=1}^B)$ $\quad\quad\quad\quad\quad\quad\quad\quad\quad\quad\quad$ ▷ Interchangeable with Sinkhorn
4: $\quad t \sim \mathcal{U}(0, 1)$
5: $\quad \{(x_0, x_1), (y_0, y_1)\}_{b=1}^B \sim \pi(z, w)$ $\quad\quad\quad\quad\quad\quad\quad\quad\quad\quad$ ▷ Resample from OT coupling
6: $\quad z = \left[(x_0^{(1)}, x_1^{(1)}), \ldots, (x_0^{(B)}, x_1^{(B)})\right], \quad w = \left[(y_0^{(1)}, y_1^{(1)}), \ldots, (y_0^{(B)}, y_1^{(B)})\right]$
7: $\quad \alpha(w) \leftarrow \exp(-\|y_0 - y_1\|_2^2 / 2\sigma_y^2)$ $\quad\quad\quad\quad\quad\quad\quad\quad$ ▷ Evaluate conditioning mismatch kernel
8: $\quad$ **if** ODE (CVFM) **then**
9: $\quad\quad p_t(x|z) \leftarrow \mathcal{N}(x; tx_1 + (1-t)x_0, \sigma_x^2)$
10: $\quad\quad p_t(y|w) \leftarrow \mathcal{N}(y; ty_1 + (1-t)y_0, \sigma_y^2)$
11: $\quad\quad x_t \sim p_t(x|z)$
12: $\quad\quad y_t \sim p_t(y|w)$
13: $\quad\quad u_t(x|z) \leftarrow x_1 - x_0$
14: $\quad\quad \mathcal{L}(\theta) \leftarrow \mathbb{E}_{t,q,p_t}\left[\alpha(w)\|v_\theta(x, y, t) - u_t(x|z)\|^2\right]$
15: $\quad$ **else if** SDE (CVSFM) **then**
16: $\quad\quad p_t(x|z) \leftarrow \mathcal{N}(x; tx_1 + (1-t)x_0, \sigma_x^2 t(1-t))$
17: $\quad\quad p_t(y|w) \leftarrow \mathcal{N}(y; ty_1 + (1-t)y_0, \sigma_y^2 t(1-t))$
18: $\quad\quad x_t \sim p_t(x|z)$
19: $\quad\quad y_t \sim p_t(y|w)$
20: $\quad\quad \hat{u}_t(x|z) \leftarrow ((1-2t)/t(1-t))(x - (tx_1 + (1-t)x_0)) + (x_1 - x_0)$
21: $\quad\quad \nabla_x \log p_t(x|z) \leftarrow (tx_1 + (1-t)x_0 - x)/(\sigma_x^2 t(1-t))$
22: $\quad\quad \mathcal{L}(\theta) \leftarrow \mathbb{E}_{t,q,p_t}\left[\alpha(w)\left(\|v_\theta(x, y, t) - \hat{u}_t(x|z)\|^2 + \lambda(t)^2\|s_\theta(x, y, t) - \nabla_x \log p_t(x|z)\|^2\right)\right]$
23: $\quad$ **end if**
24: $\quad \theta \leftarrow \text{Update}(\theta, \nabla_\theta \mathcal{L}(\theta))$
25: **end while**
26: **return** $v_\theta$ ($s_\theta$ if SDE)

---

## 4 Related Work

**Flow Matching and Schrödinger Bridges**: Simulation-free training paradigms for continuous normalizing flows (CNFs) and stochastic differential equations (SDEs) have rapidly advanced generative modeling. FM (Lipman et al., 2023; Albergo & Vanden-Eijnden, 2023; Liu et al., 2022b) provides a robust framework for regressing vector fields that transport simple base distributions to complex targets. Prior to simulation-free formulations, Diffusion Schrödinger Bridges (DSBs) (De Bortoli et al., 2021; Liu et al., 2022a; Chen et al., 2023) provided principled, albeit computationally expensive, iterative approaches to modeling stochastic dynamics, with notable applications in single-cell trajectory modeling (Bunne et al., 2023a). More recently, this framework has been elegantly extended to approximate Schrödinger Bridges via simulation-free Score and Flow Matching (Tong et al., 2023a; Shi et al., 2023), enabling the highly efficient simulation of stochastic dynamics. While highly successful for unconditional generation, scientific applications frequently require modeling dynamics subject to interventions or environmental variables, necessitating conditional frameworks.

**Conditional Optimal Transport and Dynamics**: Initial efforts to integrate conditional information into FM (Ho & Salimans, 2022; Zheng et al., 2023; Dao et al., 2023; Isobe et al., 2024) often assumed access to paired source-target samples or dense coverage of discrete conditioning classes. Recent works have pushed the boundary of FM to more complex biological and physical systems. For example, Rohbeck et al. (2024) proposed a multi-marginal FM framework to model complex cellular dynamics across multiple time points under varied chemical perturbations. However, this approach similarly assumes discrete, categorical

conditions. While not initially their primary goal, concurrent works have arisen to handle arbitrary, unpaired conditional data. These COT-FM approaches (Kerrigan et al., 2024; Chemseddine et al., 2024; Baptista et al., 2024; Fishman et al., 2026), rely instead on anisotropic ground costs to penalize transport across the conditioning dimension. While theoretically sound, as we demonstrate in our analysis, naïve mini-batch approximations of these couplings suffer from severe variance explosion in the sparse data regimes typical of continuous scientific parameters.

**Generative Modeling over Distributions**: A distinct but philosophically related line of research has formalized the problem of learning flows directly on the space of distributions. Meta Flow Matching (MFM) (Atanackovic et al., 2024) and Wasserstein Flow Matching (WFM) (Haviv et al., 2024) lift the standard flow matching objective to the Wasserstein manifold. In these frameworks, the state space is macroscopic; a single sample is an entire probability measure, represented as an empirical point cloud $X \in \mathbb{R}^{N \times d}$, and the model utilizes set-equivariant architectures to map one point cloud to another. While MFM introduces a general form of conditional generative flow matching, and WFM successfully applies this measure-wise transport to unpaired conditional distributions, these macroscopic frameworks are fundamentally incompatible with continuous conditioning spaces. In continuous settings, every unique condition $y$ is associated with a single observation. This renders set-based optimal transport and attention mechanisms ill-defined without resorting to arbitrary discretizations or binning of the continuous variable.

## 5 Experiments

We experimentally evaluate CVFM on three 2D toy problems to baseline performance and analyze training stability. Subsequently, we model the dynamics of materials' microstructures undergoing spinodal decomposition. A third case study evaluating discrete image-to-image domain transfer, alongside further experimental details and ablations, is provided in the Appendices.

**2D Experiments**: We evaluate our method on three 2D toy problems. Two problems with 8 discrete conditions – *8 Gaussians – Moons* testing large bifurcations and *8 Gaussians – 8 Gaussians* testing strong conditional dependence – are used for comparison against prior methods. A third problem, *Moons – Moons*, involves continuous conditioning (Figure E.1). CVFM is benchmarked against Conditional Generative Flow Matching (CGFM) (Isobe et al., 2024; Zheng et al., 2023; Dao et al., 2023), Wasserstein Flow Matching (WFM) (Haviv et al., 2024), Triangular Conditional Optimal Transport (T-COT-FM) (Kerrigan et al., 2024; Chemseddine et al., 2024), as well as two ablated CVFM variants. CGFM and WFM in particular do not support *unpaired* conditioning, requiring the ability to sample from $z \sim \{\pi(x_0, x_1|y_i)\}_{i=1}^m$, and precluding their ability to address scientific questions as $m \to \infty$. We simply present these methods as useful benchmarks, representing a lower bound in cases with discrete conditioning. The first ablated variant removes both mini-batch conditional OT and $\alpha(w)$, but retains the interpolated flow $p_t(y|w)$. We refer to this variant as Conditional Flow Matching (CFM) as it reduces to the originally proposed CFM given paired samples (Lipman et al., 2023; Albergo & Vanden-Eijnden, 2023). The second variant solely excludes $\alpha(w)$ – we refer to this as Conditional Optimal Transport Flow Matching (COT-FM) – nearly equivalent to T-COT-FM; it differs by having varied noise schedules for the conditional probability paths $p_t(x|z)$ and $p_t(y|w)$. ODE and SDE-based variants for the SB problem are also evaluated for CVFM and COT-FM, utilizing the probability flow ODE with entropic regularized OT, alongside the SDE extension to our objective, conditional variable score and flow matching (CVSFM).

Three mappings are investigated and their associated results are displayed in Table 1, with the Wasserstein-2 error, Maximum Mean Discrepancy (MMD), and Energy Distance (ED) from the predicted distribution at $t = 1$ to the target distribution reported. The CVFM method notably results in equivalent or better error metrics across all mappings with unpaired conditioning, nearly matching the performance of CGFM/WFM on discrete problems where these methods have knowledge of the correct conditional couplings *a priori*. Figure 3 contrasts samples and pathways from a subset of these methods. Notably, the kernel appreciably improves performance; CVFM outperforms both T-COT-FM and COT-FM, while the CFM formulation highlights the importance of the sampling from the conditional identity coupling: its absence precludes the validity of Eq. (6) (Appendix A) resulting in poor performance. Furthermore, utilizing the conditioning mismatch kernel in isolation (CVFM ($\alpha(w)$)) bypasses the computational bottleneck of mini-batch OT solvers entirely.

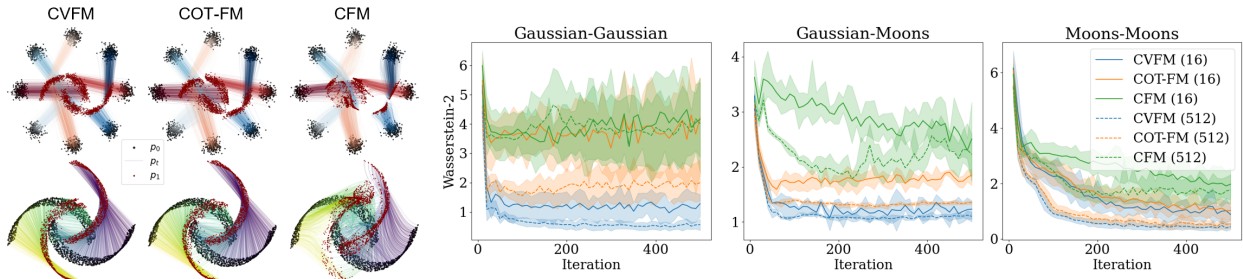

Figure 3. CVFM results in lower error in Wasserstein-2 distance to target distribution across batch sizes and conditional cost weighting $\eta$, compared to COT-FM (Eq. (9)) and the naïve conditional implementation CFM. Trajectories are colored by conditioning variable.

Table 1. Comparison of Wasserstein-2 error, Energy Distance (ED), and Maximum Mean Discrepancy (MMD) across conditional neural optimal transport and Schrödinger bridge methods. Metrics are computed between the target distribution and simulated distribution at $t = 1$. Training was repeated over 5 seeds, with values reported as $\mu \pm \sigma$. Best observed values with *unpaired samples* are bolded, with second best denoted by an asterisk within OT and SB methods. *Entropic* OT couplings identified via the Sinkhorn algorithm are differentiated from default *Exact* couplings, Eq. (3).

| | 8 Gaussians – Moons | | | 8 Gaussians – 8 Gaussians | | | Moons – Moons | | |
|---|---|---|---|---|---|---|---|---|---|
| Method | W2 (↓) | ED (↓) | MMD (↓) | W2 (↓) | ED (↓) | MMD (↓) | W2 (↓) | ED (↓) | MMD (↓) |
| CVFM | **0.246±0.013** | **0.041±0.007** | **0.037±0.004** | **0.302±0.022** | **0.056±0.012** | **0.036±0.006** | **1.101±0.054** | **0.400±0.031** | **0.407±0.022** |
| CVFM ($\alpha(w)$) | 0.304±0.056* | 0.077±0.036* | 0.057±0.020* | 0.311±0.036* | 0.057±0.020* | 0.036±0.010* | 1.145±0.042* | 0.674±0.064 | 0.539±0.026 |
| COT-FM | 0.408±0.066 | 0.122±0.054 | 0.082±0.031 | 2.106±0.182 | 0.313±0.076 | 0.070±0.012 | 1.275±0.041 | 0.511±0.058 | 0.470±0.036 |
| CFM | 1.021±0.060 | 0.803±0.095 | 0.468±0.054 | 4.549±0.138 | 3.396±0.161 | 0.571±0.017 | 2.807±0.144 | 2.917±0.239 | 1.114±0.062 |
| T-COT-FM | 0.400±0.025 | 0.083±0.020 | 0.058±0.010 | 2.029±0.169 | 0.262±0.056 | 0.058±0.010 | 1.273±0.051 | 0.462±0.048* | 0.442±0.035* |
| CVFM-Entropic | 0.321±0.045* | 0.084±0.024* | 0.060±0.011* | 0.337±0.044* | 0.074±0.025* | 0.046±0.013* | 1.203±0.046* | 0.729±0.051 | 0.568±0.025 |
| COT-FM-Entropic | 0.826±0.027 | 0.592±0.061 | 0.378±0.021 | 4.088±0.447 | 2.712±0.675 | 0.458±0.090 | 2.379±0.262 | 2.205±0.523 | 0.962±0.123 |
| CVSFM | **0.307±0.030** | **0.073±0.018** | **0.053±0.009** | **0.325±0.030** | **0.067±0.016** | **0.042±0.009** | **1.021±0.022** | **0.356±0.079** | **0.311±0.052** |
| COT-SFM | 0.404±0.048 | 0.112±0.035 | 0.072±0.017 | 2.105±0.086 | 0.326±0.078 | 0.075±0.009 | 1.221±0.048 | 0.439±0.049* | 0.349±0.031* |
| WFM | 0.369±0.094 | 0.052±0.113 | 0.064±0.075 | 0.265±0.061 | 0.062±0.028 | 0.032±0.014 | – | – | – |
| WFM-Entropic | 0.300±0.051 | 0.111±0.057 | 0.104±0.038 | 0.296±0.034 | 0.066±0.017 | 0.035±0.009 | – | – | – |
| CGFM | 0.190±0.015 | 0.025±0.006 | 0.028±0.004 | 0.228±0.017 | 0.021±0.005 | 0.018±0.003 | – | – | – |
| CGFM-Entropic | 0.219±0.011 | 0.033±0.006 | 0.033±0.003 | 0.265±0.011 | 0.034±0.004 | 0.025±0.003 | – | – | – |

As detailed in Appendix G, this lightweight variant achieves competitive performance while providing vastly superior computational scaling properties across batch sizes.

**Improved convergence**: CVFM also displays improved convergence characteristics compared with all methods directly applicable to modeling flows across unpaired conditional distributions, as evidenced by Figure 3. In all cases, the CVFM objective converges faster and to a lower metric value than other methods and variants. Further interrogation regarding the impact of $\eta$ in the conditional ground cost and $\alpha(w)$ was also performed. In the proposed methods incorporating the ground cost (CVFM, COT-FM, and T-COT-FM), increasing $\eta$ leads to improvement. However, it is often insufficient. For example, in the *8 Gaussians – 8 Gaussians* example the addition of $\alpha(w)$ in CVFM facilitates a marked improvement. Additional ablations are also presented in Appendix E.1 highlighting the important role of $\alpha(w)$ in reducing the error in the mini-batch COT approximation inherent in Eq. (9) in greater depth.

**Material Dynamics**: We next investigate the performance of our proposed method in a target scientific application: the dynamics of time-evolving microstructures subject to various processing conditions. Phase-field simulations are commonly applied to model a number of manufacturing processes involving evolving interfaces (e.g., solidification/melting, spinodal decomposition, grain growth, recrystallization, and crack propagation (Steinbach, 2009; Miehe et al., 2010)). These simulations are stochastic – the result of numerical noise representing thermal fluctuations, such that, even provided equivalent initial processing conditions, the resulting microstructure state over time will vary. In this experiment, we particularly focus on spinodal decomposition, applicable for material systems which undergo a thermodynamic phase transition resulting

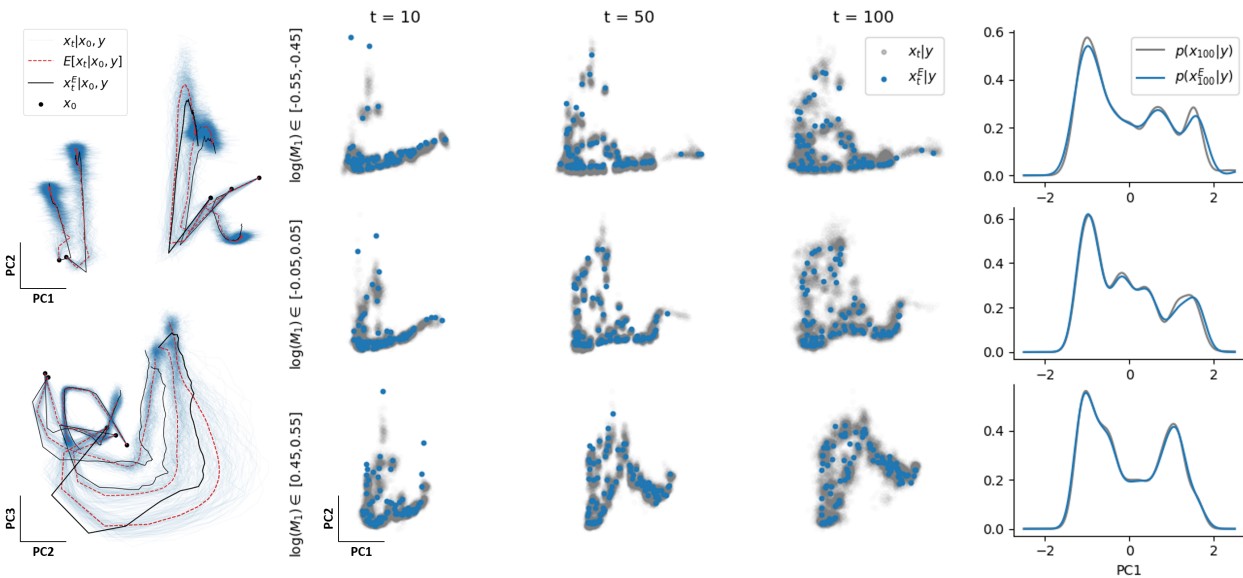

Figure 4. Collection of 5 randomly sampled trajectories from the test set in projections of PC1-PC2 and PC2-PC3 displaying (left) samples in blue from CVSFM-Exact, with the expected value as a function of time in red. Comparison of test marginal densities (middle) constructed as narrow subsets of first constituent's mobility ($\mathcal{M}_1$) parameter, in PC1-PC2 projection, and (right) marginal comparison of final state at $t = 100$ for PC1. 128 samples were simulated using the trained CVSFM-Exact predictor for each $(x_0, y)$.

in the separation of a homogeneous mixture into distinct phases. The resulting dataset contains 10,000 simulation trajectories evaluated across 100 time steps, with each frame a two-phase microstructure of dimension $256 \times 256$ voxels. Quantification of the internal state of each microstructure was performed by computing its 2-point spatial correlations (Kröner, 1971; Torquato, 2002), a representation quantifying material constituent internal spatial correlations, while also accounting for underlying symmetries (e.g., translation-equivariance and periodicity). Further discussion regarding this quantification procedure is presented in Appendix C. Individual correlations were then compressed to a 5-dimensional vector via Principal Component Analysis (PCA) defining the state variables for training. In this problem, we use CVFM to model conditional time-dependent microstructural evolution in this 5-dimensional space, provided 100 sequential conditional empirical distributions.

Importantly, the time-dependent observations modeled are the result of computational simulations rather than high-cost intermittent experimental sampling. This grants us complete access to *paired* conditioning-trajectory information that we can use for testing as well as training baseline models which are incompatible with CVFM's unpaired setting (e.g., Neural ODEs and LSTMs). In testing, we use our access to the complete paired trajectory to measure and report errors on a per trajectory basis. Specifically, model predictions start at $t = 0$ and are rolled out to predict the entire trajectory. The error reported in, for example, Table 2 is computed between this full prediction and the ground truth. Despite this, we still implement CVFM using *unpaired* data to mirror our motivating experimental settings. Table 2 contrasts the trajectory absolute error of various potential surrogates for the dynamics. We evaluate CVSFM – the SDE extension to CVFM – and alternate neural OT methods – using *unpaired* samples and conditioning variables – against traditional approaches (Neural ODEs and LSTMs) requiring complete trajectory information. To ensure fair comparison, Neural ODE and LSTM baselines were tuned to have approximately equivalent parameter counts to CVFM; see Appendix D.4 for details. Despite only having access to corrupted and *misaligned* versions of the available observations, CVSFM-Exact results in lower mean absolute errors across expected trajectories. CVSFM-Exact also outperforms alternatives such as T-COT-FM (Kerrigan et al., 2024) and an SDE-based extension, denoted as T-COT-SFM. The improvement in performance of CVSFM over paired baselines can be attributed to the specific nature of spinodal decomposition dynamics. The Cahn-Hilliard equation exhibits stiff dynamics, characterized by rapid high-gradient phase separation in early time steps followed

Table 2. Comparison of conditional neural optimal transport methods alongside conventional approaches on spinodal decomposition PC trajectories. Expected values, $\mathbb{E}[x_t|x_0, y]$, are used for stochastic predictions, initialized at $t = 0$ and rolled out until $t = 100$. Reported mean absolute errors ($\mu \pm \sigma$, minimum, and maximum) across all times.

| | Train (MAE) | | | Test (MAE) | | |
|---|---|---|---|---|---|---|
| | $\mu \pm \sigma$ | Min. | Max. | $\mu \pm \sigma$ | Min. | Max. |
| Neural ODE | 0.261±0.119 | 0.105 | **1.499** | 0.264±0.129 | 0.116 | 1.400 |
| LSTM | 0.355±0.212 | 0.038 | 2.071 | 0.358±0.331 | 0.045 | 1.648 |
| CVSFM | **0.166±0.140** | **0.023** | 1.654 | **0.188±0.147** | 0.039 | **1.284** |
| CVSFM-Entropic | 0.378±0.322 | 0.045 | 2.112 | 0.388±0.320 | 0.044 | 2.032 |
| COT-SFM | 0.202±0.348 | 0.034 | 11.422 | 0.218±0.489 | **0.033** | 11.649 |
| COT-SFM-Entropic | 0.524±0.306 | 0.108 | 2.568 | 0.531±0.307 | 0.110 | 2.012 |
| T-COT-FM | 0.899±0.370 | 0.163 | 3.470 | 0.901±0.365 | 0.161 | 2.639 |
| T-COT-SFM | 0.636±0.279 | 0.235 | 2.331 | 0.637±0.282 | 0.194 | 2.335 |

Table 3. Time-averaged Wasserstein-2 error ($\downarrow$) between predicted marginal distributions and test data for binned mobility $\log(\mathcal{M}_1)$ values.

| | $[-0.55, -0.45]$ | $[-0.05, 0.05]$ | $[0.45, 0.55]$ |
|---|---|---|---|
| CVSFM | **0.637** | 0.628 | **0.646** |
| CVSFM-Entropic | 0.886 | 1.213 | 1.523 |
| COT-SFM | 0.674 | **0.608** | 0.690 |
| COT-SFM-Entropic | 1.129 | 1.329 | 1.997 |

by slower coarsening (e.g., see Figure 4). The deterministic baseline models (i.e., Neural ODE, LSTM) are highly sensitive to approximation errors in these early high-gradient regimes; slight deviations cause the trajectory to diverge from the data manifold, leading to compounding errors across subsequent time points. Similarly, the deterministic CVFM algorithm struggled significantly in this setting. The SDE formulation of CVFM (CVSFM) incorporates a learned score function, $\nabla \log p(x|y)$, which acts as a restorative field. During sampling, this score term effectively guides the trajectory back towards high-density regions of the probability path, correcting for numerical drift caused by the stiff early dynamics and allowing CVSFM to maintain physical fidelity over longer time horizons.

We next examine CVFM's ability to accurately learn the stochasticity in the process dynamics of the microstructure in a distributional sense. Because the conditioning variables are continuous, we only have one unique trajectory $\{(x_i, y_i)\}_{i=0}^{T=99}$ for each given $y$. To overcome this limitation, we defined three narrow subsets based on the first mobility parameter $\mathcal{M}_1$, covering the test dataset. The remaining parameters $(\mathcal{M}_2, c_0)$ are not constrained. Table 3 presents the time-averaged Wasserstein-2 errors across all observed marginals for variants of CVSFM and COT-SFM on these subsets. CVSFM accurately reconstructs the marginal distributions. Figure 4 displays sample trajectories, along with the evolving time distributions for these subsets, and their corresponding marginalized density estimates along the first principal component for CVSFM. The network captures the change in the width and shape of the distribution over time. Overall, the performance on individual trajectories and observed marginals highlights the proposed method's effectiveness in capturing complex conditional stochastic nonlinear dynamics.

## 5.1 Limitations

While CVFM significantly improves stability in sparse regimes, we emphasize important limitations. First, the method introduces the kernel bandwidth $\sigma_y$ as an additional hyperparameter; while our ablations demonstrate robustness across a range of values, extreme settings can reintroduce variance or overly restrict the flow. Second, our kernel currently utilizes a Euclidean distance metric on the conditioning space. In extremely high-dimensional conditioning settings, or under non-Euclidean conditioning geometries, this

metric degrades, potentially requiring learned or domain-specific metrics. Third, further analysis in high-dimensional settings is necessary. We found that batch size and data limitations made training in these settings unstable for the unpaired frameworks. Finally, our theoretical derivation assumes that the marginal distribution over the conditioning variable remains constant across time ($q(y_0) \approx q(y_1)$), a condition satisfied in many scientific manufacturing processes but not strictly guaranteed in general.

## 6 Conclusion

We have proposed Conditional Variable Flow Matching (CVFM), a framework capable of learning to transform conditional distributions between general source and target distributions given *unpaired* samples. Building on this foundation, we also present extensions capable of approximating the conditional Schrödinger bridge problem. Our central contribution is an algorithmic framework equipped with a regularized conditional distance metric, an independent conditioning variable flow, and a conditioning mismatch kernel that enforces *local consistency* across the learned vector fields. We verify our approach through synthetic and real-world tasks, demonstrating a robust, simulation-free framework for learning conditional stochastic dynamical processes even in regimes where standard methods suffer from instabilities.

### Acknowledgments

The authors acknowledge funding by various sources. Adam P. Generale acknowledges funding from Multiscale AI and Pratt & Whitney. Surya R. Kalidindi acknowledges funding from ONR N00014-18-1-2879 as well as all previously listed funding sources. This work utilized computing resources and services provided by the Partnership for an Advanced Computing Environment (PACE) at the Georgia Institute of Technology, Atlanta, Georgia, USA. Andreas E. Robertson acknowledges the Jack Kent Cooke Foundation. In addition, this article has been authored by an employee of National Technology & Engineering Solutions of Sandia, LLC under Contract No. DE-NA0003525 with the U.S. Department of Energy (DOE). The employee owns all right, title, and interest in and to the article and is solely responsible for its contents. The United States Government retains and the publisher, by accepting the article for publication, acknowledges that the United States Government retains a non-exclusive, paid-up, irrevocable, world-wide license to publish or reproduce the published form of this article or allow others to do so, for United States Government purposes. The DOE will provide public access to these results of federally sponsored research in accordance with the DOE Public Access Plan `www.energy.gov/downloads/doe-public-access-plan`.

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

## Appendices

# A   Proofs of Theorems

## A.1   Proof $p_t(x|y)$ and $u_t(x|y)$ satisfy the continuity equation

**Theorem 3.1** *The marginal conditional vector field Eq. (6) generates the marginal conditional probability path Eq. (5) from $p_0(x|y)$ given samples of $q(z,w)$ if $q(y_0) = q(y_1)$ and the conditioning variable pairs $(y_0, y_1)$ drawn from $q(z,w)$ follow the identity coupling.*

*Proof.* The continuity equation provides a necessary and sufficient condition for a vector field to generate a probability distribution (Villani, 2009). Therefore, the proof is completed by demonstrating that $u_t(x|y)$, defined by Eq. (6), meets the continuity equation for the conditional distribution $p_t(x|y)$. We utilize the assumed decomposition introduced in the main body of the paper, $p(x, y|z, w) = p(x|z)p(y|w)$. Additionally, we assume that $x$ is independent of $w$ given $y$, $p_t(x|y, w) = p_t(x|y)$. Finally, we will require that $q(y_0) = q(y_1)$. In other words, that the distribution on the conditioning variable is the same at $t_0$ and $t_1$. We argue that this is not a very restrictive requirement in practice. It is the situation that occurs in many applications, in particular scientific dynamics problems where the distribution on conditions is constant over all time.

$$\frac{d}{dt}\left(p_t(x|y)\right) = -\text{div}_x(p_t(x|y)u_t(x|y)) \tag{A.1}$$

We begin with expanding the left hand side of the continuity equation in Eq. (A.1). To clarify notation, we utilize $p(\cdot)$ to denote probability density functions over the primary variables, $x$ and $y$. We use $q(\cdot)$ to denote distributions over the pair variables $z$ and $w$. The subscript $t$ denotes time dependence. Their arguments differentiate the multiple distributions of each type.

$$\frac{d}{dt}\left(p_t(x|y)\right) = \frac{d}{dt}\left(\frac{p_t(x,y)}{p_t(y)}\right)$$
$$= \frac{d}{dt}\left(\int \frac{p_t(x,y|w,z)}{p_t(y)}q(w,z)dzdw\right)$$

We utilize the following assumed decomposition, $p_t(x, y|w, z) = p_t(x|z)p_t(y|w)$,

$$= \frac{d}{dt}\left(\int \frac{p_t(x|z)p_t(y|w)}{p_t(y)}q(w,z)dzdw\right)$$

$$= \int \left[\underbrace{\frac{p_t(y|w)}{p_t(y)}\frac{d}{dt}(p_t(x|z))}_{T_1} - \underbrace{\frac{p_t(x|z)p_t(y|w)}{p_t(y)^2}\frac{d}{dt}(p_t(y))}_{T_2} + \underbrace{\frac{p_t(x|z)}{p_t(y)}\frac{d}{dt}(p_t(y|w))}_{T_3}\right]q(w,z)dzdw$$

We next consider each of the individual terms above. We will extensively rely upon the fact that we have prescribed forms to $u_t(x|z)$ and $u_t(y|w)$ such that they generate $p_t(x|z)$ and $p_t(y|w)$, respectively (see Theorem 3 of Lipman et al. (2023) or Theorem 2.1 in Tong et al. (2023b)). The integrands are further assumed to satisfy the regularity conditions of the Leibniz Rule for changing the order of integration and differentiation. Beginning with the first term, $T_1$,

$$T_1 = \int \frac{p_t(y|w)}{p_t(y)}\frac{d}{dt}(p_t(x|z))q(w,z)dzdw$$

as $u(x|z)$ generates $p(x|z)$,

$$= -\int \frac{p_t(y|w)}{p_t(y)} \mathrm{div}_x(p_t(x|z)u_t(x|z))q(w,z)dzdw$$

Leibniz Rule,

$$= -\mathrm{div}_x\left(\int \frac{p_t(y|w)p_t(x|z)q(w,z)}{p_t(y)}u_t(x|z)dzdw\right)$$

$$= -\mathrm{div}_x\left(p_t(x|y)\int \frac{p_t(y|w)p_t(x|z)q(w,z)}{p_t(y)p_t(x|y)}u_t(x|z)dzdw\right)$$

using the definition of $u_t(x|y)$, Eq. (6),

$$= -\mathrm{div}_x\left(p_t(x|y)u_t(x|y)\right)$$

Here, we arrive at the right hand side of the continuity equation. Next, we turn to demonstrating that terms $T_2$ and $T_3$ equate to zero. We begin inspecting the second term, $T_2$,

$$T_2 = \int -\frac{p_t(x|z)p_t(y|w)}{p_t(y)^2}\frac{d}{dt}(p_t(y))q(w,z)dzdw$$

$$= -\frac{1}{p_t(y)^2}\frac{d}{dt}(p_t(y))\int p_t(x|z)p_t(y|w)q(w,z)dzdw$$

$$= -\frac{p_t(x|y)}{p_t(y)}\frac{d}{dt}p_t(y)$$

Next, we consider the last term, $T_3$.

$$T_3 = \int \frac{p_t(x|z)}{p_t(y)}\frac{d}{dt}(p_t(y|w))q(w,z)dzdw$$

as $u_t(y|w)$ generates $p_t(y|w)$,

$$= -\int \frac{p_t(x|z)}{p_t(y)}\mathrm{div}_y(p_t(y|w)u_t(y|w))q(w,z)dzdw$$

Leibniz Rule,

$$= -\frac{1}{p_t(y)}\mathrm{div}_y\left(\int u_t(y|w)p_t(x|z)p_t(y|w)q(w,z)dzdw\right)$$

marginalization on $z$, the chain rule of probability, and the second assumption outlined previously,

$$= -\frac{1}{p_t(y)}\mathrm{div}_y\left(\int u_t(y|w)p_t(x|y)p_t(y|w)q(w)dw\right)$$

$$= -\frac{1}{p_t(y)}\mathrm{div}_y\left(p_t(x|y)p_t(y)\int u_t(y|w)\frac{p_t(y|w)q(w)}{p_t(y)}dw\right)$$

definition of the marginal vector field, Eq. (8) in Lipman et al. (2023) or Eq. (9) in Tong et al. (2023b)

$$= -\frac{1}{p_t(y)}\mathrm{div}_y\left(p_t(x|y)p_t(y)u_t(y)\right)$$

product rule,

$$= -\frac{1}{p_t(y)}p_t(y)u_t(y)^T\nabla_y p_t(x|y) - \frac{p_t(x|y)}{p_t(y)}\mathrm{div}_y\left(p_t(y)u_t(y)\right)$$

$u_t(y)$ generates $p_t(y)$,

$$= -\frac{1}{p_t(y)}p_t(y)u_t(y)^T\nabla_y p_t(x|y) + \frac{p_t(x|y)}{p_t(y)}\frac{d}{dt}p_t(y)$$

$$= -u_t(y)^T\nabla_y p_t(x|y) + \frac{p_t(x|y)}{p_t(y)}\frac{d}{dt}p_t(y)$$

Combining $T_1$, $T_2$, and $T_3$ together, we obtain,

$$\frac{d}{dt}\left(p_t(x|y)\right) = T_1 + T_2 + T_3$$

$$= -\mathrm{div}_x\left(p_t(x|y)u_t(x|y)\right) - \frac{p_t(x|y)}{p_t(y)}\frac{d}{dt}p_t(y) - u_t(y)^T\nabla_y p_t(x|y) + \frac{p_t(x|y)}{p_t(y)}\frac{d}{dt}p_t(y)$$

$$= -\mathrm{div}_x\left(p_t(x|y)u_t(x|y)\right) - u_t(y)^T\nabla_y p_t(x|y)$$

**Vanishing of the residual term under the identity coupling.** The continuity equation is satisfied if and only if the residual $u_t(y)^\top\nabla_y p_t(x|y)$ vanishes. We now argue that under the assumptions of CVFM, this residual is zero.

The marginal flow $u_t(y)$ is constructed by marginalizing the conditional flow $u_t(y|w)$ over $q(w)$, in direct analogy to Lipman et al. (2023):

$$u_t(y) = \int u_t(y|w)\frac{p_t(y|w)q(w)}{p_t(y)}\,dw. \tag{A.2}$$

For the prescribed Gaussian conditional path $p_t(y|w) = \mathcal{N}(y; ty_1 + (1-t)y_0, \sigma_y^2)$ used in CVFM, the per-pair conditional flow is $u_t(y|w) = y_1 - y_0$, which is generically nonzero.

However, when $q(y_0) = q(y_1)$ as underlying distributions, the optimal transport plan between them is the identity coupling, under which $y_0 = y_1$ almost surely and hence $u_t(y|w) = 0$ almost surely. Marginalizing then gives $u_t(y) = 0$ for all $t \in [0, 1]$.

In practice, CVFM does not directly sample from the identity coupling – rather, it approximates it through (i) static mini-batch OT under the anisotropic cost $c_\eta$ of Eq. (9), which converges to the identity coupling as $\eta \to \infty$ by Proposition 3.1, and (ii) the conditioning mismatch kernel $\alpha(w)$, which down-weights residual mismatched pairings (Theorem 3.2). Following the analogous arguments in Tong et al. (2023b) and Pooladian et al. (2023), the expectation of the conditional flow under this approximate coupling converges to the marginal OT flow, namely, zero. Therefore, $u_t(y) \approx 0$ in expectation, and the residual term $u_t(y)^\top\nabla_y p_t(x|y)$ vanishes in expectation under CVFM training, satisfying the continuity equation

$$\frac{d}{dt}\left(p_t(x|y)\right) = -\mathrm{div}_x\left(p_t(x|y)u_t(x|y)\right). \tag{A.3}$$

As a consequence, without static mini-batch OT and the anisotropic cost $c_\eta$, the learned $u_t(x|y)$ cannot generate the required marginal conditional vector field generating $p_t(x|y)$. We directly observed the impact

of the residual term in the first presented case study (Table 1). Only once conditional optimal transport was incorporated could the appropriate push-forward operation be obtained. Importantly, this behavior differs from previous efforts where learning the appropriate vector field is possible with or without optimal transport resampling (Albergo et al., 2023; Pooladian et al., 2023; Liu et al., 2022b; Tong et al., 2023b).

## A.2  Equivalence of the Flow Matching Objective

**Theorem 3.2** *Let $p_t(x|y) > 0$ for all $x \in \mathbb{R}^N$, $y \in \mathbb{R}^M$, $t \in [0,1]$ and $\mathcal{L}_{MCFM}(\theta) = \mathbb{E}_{t,p_t(x,y)}\|v_\theta(x,y,t) - u_t(x|y)\|_2^2$. Let $\alpha$ be a stationary, isotropic kernel depending only on $r = \|y_0 - y_1\|_2$, decaying to $0$ as $r \to \infty$, and $\mathcal{C}^2$ in a neighborhood of $0$ with $\alpha(0) = 1$ and $\alpha'(0) = 0$, and further assume the coupling $\pi(w)$ satisfies $\mathbb{E}_{\pi(w)}[r^2] < \infty$. Then there exists a constant $C$ such that for every $\theta$,*

$$\left\|\nabla_\theta \mathcal{L}_{CVFM}(\theta) - \nabla_\theta \mathcal{L}_{MCFM}(\theta)\right\| \le C\,\mathbb{E}\left[\|y_0 - y_1\|_2^2\right]. \tag{A.4}$$

*If $\pi(w)$ enforces $\mathbb{E}_{\pi(w)}[\|y_0 - y_1\|_2^2] = O(\varepsilon^2)$ then the gradient error is $O(\varepsilon^2)$.*

*Proof.* Several assumptions are necessary to guarantee the existence of various integrals and to allow exchanging of their order. Specifically, we assume $q(w,z)$, $p_t(x|z)$, $p_t(y|w)$ and $p(w)$ decrease to zero as $\|x\|, \|y\| \to \infty$ and that $v_t$, $v_t$, and $\nabla_\theta v_t$ are bounded. Furthermore, the optimal transport penalty, $\eta$, is taken to be sufficiently large such that when sampling in mini-batches approximating the optimal coupling between $q(y_0)$ and $q(y_1)$, $y_0 \approx y_1$. Additionally, we assume $\alpha$ is $\mathcal{C}^2$ in a neighborhood of $0$ with $\alpha(0) = 1$ and $\alpha'(0) = 0$.

We begin by stating the intractable marginal conditional flow matching (MCFM) objective

$$\mathcal{L}_{MCFM}(\theta) = \mathbb{E}_{t,p_t(x,y)}\|v_\theta(x,y,t) - u_t(x|y)\|^2 \tag{A.5}$$

Expanding the $L_2$-norm

$$\mathcal{L}_{MCFM}(\theta) = \mathbb{E}_{t,p_t(x,y)}\|v_\theta(x,y,t) - u_t(x|y)\|^2$$

$$= \mathbb{E}_{t,p_t(x,y)}\left[\underbrace{\|v_\theta(x,y,t)\|^2}_{T_2} - \underbrace{2\left\langle v_\theta(x,y,t), u_t(x|y)\right\rangle}_{T_1} + \underbrace{\|u_t(x|y)\|^2}_{T_3}\right]$$

Consider each term independently. Note, the third term can be ignored because it is independent of the trainable parameters.

$$T_1 = \mathbb{E}_{t,p_t(x,y)}\left[\|v_\theta(x,y,t)\|^2\right]$$

$$= \mathbb{E}_t\left[\int p_t(x,y,w,z)\|v_\theta(x,y,t)\|^2 dz\,dw\,dx\,dy\right]$$

$$= \mathbb{E}_t\left[\int p_t(x|z)p_t(y|w)q(z,w)\|v_\theta(x,y,t)\|^2 dz\,dw\,dx\,dy\right]$$

$$= \mathbb{E}_{t,p_t(x|z),p_t(y|w),q(z,w)}\left[\|v_\theta(x,y,t)\|^2\right]$$

Consider the second term.

$$
\begin{aligned}
T_2 &= -2\mathbb{E}_{t,p_t(x,y)}\left[\langle v_\theta(x,y,t), u_t(x|y)\rangle\right] \\
&= -2\mathbb{E}_t\left[\int p_t(x,y)\left\langle v_\theta(x,y,t), \int u_t(x|z)\frac{p_t(x,y|z,w)q(z,w)}{p_t(x,y)}dzdw\right\rangle dxdy\right] \\
&= -2\mathbb{E}_t\left[\int p_t(x,y)\frac{p_t(x,y|z,w)q(z,w)}{p_t(x,y)}\langle v_\theta(x,y,t), u_t(x|z)\rangle dxdydzdw\right] \\
&= -2\mathbb{E}_t\left[\int p_t(x|z)p_t(y|w)q(z,w)\langle v_\theta(x,y,t), u_t(x|z)\rangle dxdydzdw\right] \\
&= -2\mathbb{E}_{t,p_t(x|z),p_t(y|w),q(z,w)}\left[\langle v_\theta(x,y,t), u_t(x|z)\rangle\right]
\end{aligned}
$$

Combining these together and comparing them against the expanded form for $\mathcal{L}_{\text{CVFM}}(\theta)$, we clearly see that the $\mathcal{L}_{\text{MCFM}}(\theta)$ and $\mathcal{L}_{\text{CVFM}}(\theta)$ objectives are equivalent up until a constant independent of the training parameters, $\theta$.

The above equality demonstrates equivalence in the absence of the kernel $\alpha$, where the only difference between the two objectives is the multiplicative factor $\alpha(w)$ appearing in the complete CVFM objective. As $\alpha(w)$ depends only on $w = (y_0, y_1)$ through $r = \|y_0 - y_1\|_2$, we define $\alpha(w) = \tilde{\alpha}(r)$ with $\tilde{\alpha} : \mathbb{R}_{\geq 0} \to \mathbb{R}$. Consequently the gradient difference can be written in the form

$$
\nabla_\theta \mathcal{L}_{\text{CVFM}}(\theta) - \nabla_\theta \mathcal{L}_{\text{MCFM}}(\theta) = \mathbb{E}_{q(w)}\left[(\tilde{\alpha}(r) - 1)h_\theta\right], \tag{A.6}
$$

where $h_\theta$ is the grouping of terms $T_1$ and $T_2$ from above, including the expectations over the remaining variables. By the uniform-boundedness assumption there exists $H < \infty$ with $|h_\theta| \leq H$ for all relevant $\theta$.

By hypothesis $\tilde{\alpha}$ is $\mathcal{C}^2$ in a neighborhood of 0 with $\tilde{\alpha}(0) = 1$ and $\tilde{\alpha}'(0) = 0$, so a second-order Taylor expansion about $r = 0$ yields

$$
\tilde{\alpha}(r) - 1 = \tfrac{1}{2}\tilde{\alpha}''(0)r^2 + \rho(r), \qquad \rho(r) = o(r^2) \text{ as } r \to 0. \tag{A.7}
$$

Substituting this expansion into the gradient identity and taking norms gives the bound

$$
\|\nabla_\theta \mathcal{L}_{\text{CVFM}}(\theta) - \nabla_\theta \mathcal{L}_{\text{MCFM}}(\theta)\| \leq H\,\mathbb{E}_{q(w)}\left[|\tilde{\alpha}(r) - 1|\right] \leq H\left(\tfrac{1}{2}|\tilde{\alpha}''(0)|\,\mathbb{E}_{q(w)}[r^2] + \mathbb{E}_{q(w)}[|\rho(r)|]\right). \tag{A.8}
$$

Since $\rho(r) = o(r^2)$ is integrable pointwise, dominated convergence implies $\mathbb{E}_{q(w)}[|\rho(r)|] = o(\mathbb{E}_{q(w)}[r^2])$. Hence the right-hand side is $O(\mathbb{E}_{q(w)}[r^2])$, so there exists a constant $C$ (depending only on $H$ and $\tilde{\alpha}''(0)$) with

$$
\|\nabla_\theta \mathcal{L}_{\text{CVFM}}(\theta) - \nabla_\theta \mathcal{L}_{\text{MCFM}}(\theta)\| \leq C\mathbb{E}_{q(w)}[r^2], \tag{A.9}
$$

which is the claimed inequality. The refined identity

$$
\nabla_\theta \mathcal{L}_{\text{CVFM}}(\theta) - \nabla_\theta \mathcal{L}_{\text{MCFM}}(\theta) = \tfrac{1}{2}\tilde{\alpha}''(0)\mathbb{E}_{q(w)}[r^2 h_\theta] + o(\mathbb{E}_{q(w)}[r^2]) \tag{A.10}
$$

shows that the linear term vanishes and the leading contribution is quadratic in $r$, so the approximation is exact to second order. Finally, if $\tilde{\alpha} \equiv 1$ then $\tilde{\alpha}(r) - 1 \equiv 0$ and the gradient difference vanishes identically.

### A.3 Continuity of Conditional Flows

**Theorem 3.3** *Let the learned vector field $v_\theta(x, y, t)$ be locally Lipschitz continuous with respect to both $x$ and $y$. Let $\phi_t(x_0, y)$ be the path of a particle starting at $x_0$ generated by the ODE $\frac{dx}{dt} = v_\theta(x, y, t)$. Then, for any fixed $t \in [0, 1]$ and initial distribution $p_0(x)$:*

1. *The transport map $\phi_t(x_0, \cdot)$ is continuous with respect to the conditioning variable $y$.*

2. *The pushforward probability measure $(\phi_t(\cdot, y))_{\#}p_0 = p_t(\cdot|y)$ is continuous with respect to $y$ in the Wasserstein sense.*

*Proof.* Let $y_1$ and $y_2$ be two distinct conditioning variables. The corresponding paths are solutions to the integral equations:

$$\phi_t(x_0, y_i) = x_0 + \int_0^t v_\theta(\phi_s(x_0, y_i), y_i, s) \, ds \quad \text{for } i = 1, 2 \tag{A.11}$$

By the assumption that $v_\theta$ is locally Lipschitz continuous in both $x$ and $y$, standard theorems on ODEs guarantee that the solution $\phi_t(x_0, y)$ is continuous with respect to the parameter $y$, proving the first claim.

The continuity of the pushforward measure follows from the continuity of the transport map. The squared Wasserstein-2 distance between the two pushforward measures is bounded by the expected squared distance between the paths:

$$W_2^2(p_t(\cdot|y_1), p_t(\cdot|y_2)) \leq \mathbb{E}_{x_0 \sim p_0(x)} \left[ \|\phi_t(x_0, y_1) - \phi_t(x_0, y_2)\|^2 \right] \tag{A.12}$$

Since $\phi_t(x_0, y)$ is continuous in $y$ for each $x_0$, as $y_2 \to y_1$, the right-hand side converges to 0 (assuming mild regularity conditions on $p_0$). Therefore, $p_t(\cdot|y)$ is continuous in $y$ with respect to the Wasserstein distance, proving the second claim.

# B    Additional Simulation-Free Score and Flow Matching Background

Tong et al. (2023a) proposed a simulation-free training objective for approximating continuous time Schrödinger bridges (Bunne et al., 2023a; De Bortoli et al., 2023), generalizing Flow Matching (FM) (Lipman et al., 2023; Albergo & Vanden-Eijnden, 2023) to the case of stochastic dynamics with arbitrary source distributions. Let $p : [0, 1] \times \mathbb{R}^N \to \mathbb{R}^+$ define a time-dependent probability path, $u : [0, 1] \times \mathbb{R}^N \to \mathbb{R}^N$ a time-dependent vector field, and $g : [0, 1] \to \mathbb{R}_{>0}$ a continuous positive diffusion function. An associated Itô stochastic differential equation (SDE) can be defined

$$dx = u_t(x)dt + g(t)dw_t \tag{B.1}$$

where $u_t(x)$ is equivalent to $u(t, x)$, and $dw_t$ is standard Brownian motion. Utilizing the *Fokker-Planck equation* and *continuity equation*, it is possible to derive the *probability flow* ordinary differential equation (ODE) of the process (Tong et al., 2023a; Song et al., 2021) and establish a relation between the probability flow ODE $\hat{u}_t(x)$ and the SDE drift as

$$u_t(x) = \hat{u}_t(x) + \frac{g^2(t)}{2} \nabla \log p_t(x). \tag{B.2}$$

As long as the probability flow ODE and score function can be specified, the SDE can be adequately described. Tong et al. (2023a) demonstrated that the intuition underpinning Eq. (5) can also be extended to the marginalization over conditional scores, resulting in the expressions

$$\hat{u}_t(x) = \int \hat{u}_t(x|z) \frac{p_t(x|z)q(z)}{p_t(x)} dz$$
$$\nabla \log p_t(x) = \int \nabla \log p_t(x|z) \frac{p_t(x|z)q(z)}{p_t(x)} dz. \tag{B.3}$$

**Gaussian marginal conditional flows**: Prior works (Theorem 3 (Lipman et al., 2023), Theorem 2.1 (Tong et al., 2023b), Theorem 2.6, (Albergo & Vanden-Eijnden, 2023)) have demonstrated a method for tractably evaluating Eq. (B.3) in the case where the ODE/SDE conditional flows are Gaussian (i.e., $p_t(x|z) = \mathcal{N}(x; \mu_t(z), \sigma_t^2(z))$). The unique vector field $\hat{u}_t(x|z)$ generating this flow has the form

$$\hat{u}_t(x|z) = \frac{\sigma_t'(z)}{\sigma_t(z)}(x - \mu_t(z)) + \mu_t'(z) \tag{B.4}$$

where $\sigma_t'(z)$ and $\mu_t'(z)$ denote the time derivatives of $\sigma_t(z)$ and $\mu_t(z)$, respectively. This can seamlessly be extended to define the conditional score $\nabla \log p_t(x|z) = -(x - \mu_t(z))/\sigma_t^2(z)$. In the particular case of a

*Brownian bridge* from $x_0$ to $x_1$, sampled from $q(z)$, with constant diffusion rate $g(t) = \sigma$, the conditional flow is defined as $p_t(x|z) = \mathcal{N}(x; tx_1 + (1-t)x_0, \sigma^2 t(1-t))$, resulting in

$$
\begin{aligned}
\hat{u}_t(x|z) &= \frac{1-2t}{t(1-t)}(x - (tx_1 + (1-t)x_0)) + (x_1 - x_0) \\
\nabla \log p_t(x|z) &= \frac{tx_1 + (1-t)x_0 - x}{\sigma^2 t(1-t)}.
\end{aligned}
\tag{B.5}
$$

In a similar manner to the derivation shown in Appendix A, a density over initial conditions $p(x_0)$, induces marginal distributions $p_t(x)$ satisfying the *Fokker-Planck equation* where $\Delta p_t = \nabla \cdot (\nabla p_t)$ (Tong et al., 2023a):

$$
\frac{\partial p}{\partial t} = -\nabla \cdot (p_t u_t) + \frac{g^2(t)}{2}\Delta p_t
\tag{B.6}
$$

In total, Tong et al. (2023a) demonstrate that the concept of regressing upon the conditional vector field from FM can be extended to regressing upon conditional drift and score, providing improved performance in practice.

**Weighting schedule** $\lambda(t)$: In the case with conditional Gaussian $p_t(x|z)$ probability paths, as in Eq. (B.4), Tong et al. (2023a) advocate a particular weighting schedule $\lambda(t)$:

$$
\lambda(t) = \frac{2\sigma_t}{\sigma^2} = \frac{2\sqrt{t(1-t)}}{\sigma}.
\tag{B.7}
$$

This weighting schedule provides simplification to the objective alongside numerical stability, converting the score matching objective to

$$
\lambda(t)^2 \|s_\theta(x, t) - \nabla_x \log p_t(x|z)\|^2 = \|\lambda(t)s_\theta(x, t) + \varepsilon\|^2
\tag{B.8}
$$

where $\varepsilon \sim \mathcal{N}(0, 1)$.

## C  Case Study Background

### C.1  Microstructure Evolution in Materials Informatics

In this section, we provide a brief background of relevant topics for the third case study, describing relevant materials background at a high level. Appendix D and Appendix E provide more details on the datasets and empirical distributions involved in training as well as greater analysis of the case study results.

**Phase-Field modeling**: Phase-field simulations are commonly applied to model a number of manufacturing processes involving evolving interfaces (such as solidification/melting, spinodal decomposition, grain growth, recrystallization, and crack propagation (Steinbach, 2009; Miehe et al., 2010)). In particular, the Cahn-Hilliard equation is frequently used to describe spinodal decomposition, a spontaneous thermodynamic-instability-induced phase separation (Cahn & Hilliard, 1958). This partial differential equation models a diffusion driven process which minimizes the total energy of the system by inducing phase separation. In this work, we use the formulation described in Montes de Oca Zapiain et al. (2021); we use the standard Cahn-Hilliard equation with a double well potential describing the local contribution to the total energy Dingreville et al. (2020). The material state is represented using a single, continuous field variable, $c(x)$, that separates into two phases ($c = -1$ and $c = 1$).

$$\frac{\partial c}{\partial t} = \nabla \cdot \left( \mathcal{M}(c) \nabla \frac{\partial f(c)}{\partial c} - \kappa \nabla^2 c \right) \tag{C.1a}$$

$$f(c) = \omega(c-1)^2(c+1)^2 \tag{C.1b}$$

$$\mathcal{M}(c) = s(c)\mathcal{M}_1 + \left(1 - s(c)\right)\mathcal{M}_2 \tag{C.1c}$$

$$s(c) = \frac{1}{4}(2-c)(1+c)^2 \tag{C.1d}$$

The values of the $c$ field corresponding to each separated phase are defined by the minima of the chemical free energy density, $f(c)$. The spatially-dependent concentration field, $c(x)$, will take the role of material microstructure. The concentration dependent mobility, $\mathcal{M}(c)$, which defines the diffusion dynamics of the concentration field, $c(x)$, is parameterized by $\mathcal{M}_1$ and $\mathcal{M}_2$ which are scalar constant mobility parameters for each phase. Parameters $\omega$ and $\kappa$, the energy barrier height and the gradient energy coefficient, respectively, are both set to 1. For the purpose of modeling the system's stochastic dynamics using CVFM, the neural network is conditioned on three parameters: $\log(\mathcal{M}_1)$, $\log(\mathcal{M}_2)$, and $c_0$ – the initial concentration of the $c$ field at time zero. We note that at time zero, $c(x) = c_0 + \epsilon(x)$, where $\epsilon(x)$ is a low variance, clipped white noise field (i.e., values cannot exceed $[-1, 1]$ after the sum) (e.g., $\mathrm{STD}[\epsilon(x)] = 0.35$). This stochasticity is necessary to initiate phase separation and results in a distribution of final microstructures when a single set of $\mathcal{M}_1, \mathcal{M}_2, c_0$ are rerun several times. The dataset utilized in this paper's third presented case study was simulated under periodic boundary conditions using the MEMPHIS code base from Sandia National Labs (Dingreville et al., 2020). Details on the dataset are included in Appendix D.4.

**2-Point spatial correlations**: This work uses 2-point spatial correlations (Torquato, 2002; Kalidindi, 2015; Adams et al., 2013) as descriptive microstructure features to quantify the state of the material microstructure at a given time. This featurization builds on the idea that the microstructure itself is a stochastic function, where individual observed microstructure instances are simply samples from governing stochastic microstructure function (SMF) (Kröner, 1971; Torquato, 2002; Niezgoda et al., 2011). This concept was developed and is used to account for the observed degeneracy in material microstructures represented in direct space (e.g., for spinodal decomposition, represented directly as the concentration field, $c(x)$). This degeneracy refers to the observation that visually dissimilar microstructures sampled from the same underlying SMF are characterized by identical physical properties and process evolution in response to an applied environment. Therefore, it is more physically relevant (and computationally simple) to quantify evolution in SMFs than individual observed microstructures. Within this conceptualization, the underlying stochastic microstructure function is identified from individual instances by empirically estimating the moments of the SMF – e.g., its second order moments: the 2-point statistics. In practice, it is generally sufficient to represent material microstructures using their 2-point statistics. In addition to addressing degeneracy, a 2-point statistics based representation also provides a convenient mechanism to account for underlying symmetries (such as translation-equivariance and periodicity).

In practice, 2-point spatial correlations can be computed as a convolution of the sampled discrete microstructure function $m_s^\alpha$, where $\alpha$ indexes the material local state and $s$ indexes the spatial voxel. For the considered spinodal decomposition problem, $m_s = c_s$ has a one dimensional state. The resulting 2-point spatial correlations between two arbitrary material states, $\alpha$ and $\beta$, are then defined by the operation

$$f_r^{\alpha\beta} = \frac{1}{S} \sum_{s=1}^{S} m_s^\alpha m_{s+r}^\beta \tag{C.2}$$

where $S$ is the number of voxels in the microstructural domain. Dimensionality reduction techniques can then be effectively applied to a dataset of 2-point spatial correlations to provide robust, information-dense features for the construction of linkages between process parameters and internal material structure (Gupta

et al., 2015; Latypov et al., 2019; Yabansu et al., 2020; Marshall & Kalidindi, 2021; Paulson et al., 2017; Generale & Kalidindi, 2021; Kalidindi, 2020; Harrington et al., 2022; Generale et al., 2023; 2024; Generale, 2024).

## D   Experiment Implementation Details

### D.1   Static Optimal Transport

Static exact and entropic regularized optimal transport couplings were solved for in mini-batches during training through the Python Optimal Transport (POT) package (Flamary et al., 2021) (`https://python ot.github.io/`). As similarly reported in (Tong et al., 2023a), we noticed improved performance in the low-dimensional toy cases with the Sinkhorn algorithm (Cuturi, 2013), which degraded in higher dimensions (e.g., material dynamics, domain transfer). The use of mini-batch optimal transport has also been previously shown to regularize the transport plan (Fatras et al., 2021b;a) due to the stochastic nature of the independent batch samplings forming non-optimal couplings, in effect resulting in entropic-regularized OT plans, even with exact OT solves.

### D.2   Computational Resources

All experiments were performed on a high-performance-computing cluster with CPU nodes of 24 CPUs and GPU nodes with V100 and A100 GPUs. 2D experiments were all performed on 1 V100, domain transfer and material dynamics experiments were performed on 4x V100's or 2x A100's.

### D.3   2D Experimental Details

The 2D experiments involve three toy examples: *8 Gaussians – Moons*, *8 Gaussians – 8 Gaussians*, and *Moons – Moons*. These simple case studies are meant to analyze the training stability of CVFM (and ablations of CVFM), baseline CVFM's performance against alternative solutions, and provide a sufficiently low-dimensional setting to visualize and analyze the flow pathways. The next paragraphs detail the data generation process for the toy case studies. For all three, mini-batches were generated on the fly from new samples from the base distributions. In Figure E.1, the source distributions are labeled with blue samples and the target distributions are labeled with red samples.

The *8 Gaussians – Moons* toy problem involves discrete conditioning that can take 8 individual values. Each condition value uniquely identifies one of the 8 Gaussians. The probability of each condition value was uniform. The two moons are each separated into 4 corresponding sections each. Each moon was separated by assigning a given sample to a select condition value dictated by the sample's relative angle with respect to the origin of the containing moon. Each moon was broken into 4 condition value groups based on 45º degree segments. As shown in Figure E.1, the individual Gaussians share condition value with a specific moon in an alternating pattern. This leads to a flow with greater bifurcations. During training, new samples are generated for each mini-batch. Samples from the 8 Gaussians and the Moons (including both x-location in 2D space and condition value, $y$) are generated independently to construct a mini-batch sampling from $q_{\text{Gauss}}(x_0, y_0)$ and $q_{\text{moon}}(x_1, y_1)$.

The *8 Gaussians – 8 Gaussians* toy problem also involves discrete conditioning that can take 8 unique values. The probability of each condition value was uniform. Each set of 8 Gaussians is arranged in a circle, Figure E.1. A single unique condition value is assigned to each Gaussian in a given set. The condition value assignments for each set are offset from each other by a single place rotation. Empirically, we observed that the proximity of the two rings leads to a flow with strong coupling between $x$ and $y$ that is challenging to learn. Samples are generated on the fly in each mini-batch. Each ring is sampled independently (first the Gaussian index is sampled, and then $x$-coordinates are sampled). The outer ring is the source distribution: $q_{\text{outer}}(x_0, y_0)$ and $q_{\text{inner}}(x_1, y_1)$.

Finally, the Moons-Moons toy problem involves continuous conditioning. As shown in Figure E.1, the red target moons are rotated 90º degrees with respect to the blue source moons. The rotation is performed about the combined origin of the two moons in the source distribution. Continuous conditioning is assigned

based on the absolute location of a sample. The same expression is used for both the source and target distributions: $y = (x_0 - 10)I_{[0,1]} + (1 - I_{[0,1]})(x_0 + 10)$. Here, $x_0$ refers to the 0 component of the 2D $x$ (not $x$ at time 0 as has been used in the rest of the paper). $I_{[0,1]}$ is a binary indicator denoting which moon in the pair the sample came from. To generate the target distribution, samples are drawn from the source. Then, the sample conditional label is assigned based on the reported equation. The probability of each condition was defined implicitly by the sampling process. Finally, the samples are rotated 90° degree. Via these operations, samples within the mini-batch are drawn independently from distributions $q_{M1}(x_0, y_0)$ and $q_{M2}(x_1, y_1)$.

For all 2D synthetic dataset cases we used networks of four layers with width 128 and GELU activations (Hendrycks & Gimpel, 2023). In the WFM variants, we added a permutation equivariant encoder for distributional context with encoding dimension of 16, 4 attention heads, and self-attention embedding of 32. Optimization was carried out with a constant learning rate of $1e - 3$ and ADAM-W (Loshchilov & Hutter, 2019) over 10,000 steps and a batch size of 256, unless otherwise specified. Sampling was performed by integration with the adaptive step size `dopri5` solver and tolerances `atol = rtol = 1e - 5`. Conditional probability paths $p_t(x|z)$ were defined with $\sigma_x = 0.1$, which was held constant throughout all cases. Values of $\eta$ and $\sigma_y$ for $p_t(y|w)$ were varied between discrete and continuous conditioning cases. In discrete conditioning cases (*8 Gaussians – 8 Gaussians* and *8 Gaussians – Moons*), $\sigma_y = 0.02$ and $\eta = 100$, whereas in the continuous conditioning case (*Moons – Moons*), $\sigma_y = 0.5$ and $\eta = 5$.

Empirical values of the Wasserstein-2 distance were evaluated through 2,048 samples simulated through the learned conditional vector fields and computed against an equivalent number of samples from the target distribution. The $W_2$ reported distance differs depending on cases with continuous or discrete conditioning. In the discrete case, we take the mean of the conventional $W_2$ distance across all conditioning classes

$$W_2(\hat{p}_1, q_1) = \mathbb{E}_{y \sim p(y)} \left[ \left( \inf_{\pi \in \Pi(\hat{p}_1, q_1)} \int \|x_i - x_j\|^2 d\pi(x_i, x_j) \right)^{1/2} \right] \tag{D.1}$$

while in the continuous case, we incorporate the conditional ground cost as

$$W_2(\hat{p}_1, q_1) = \left( \inf_{\pi \in \Pi(\hat{p}_1, q_1)} \int \left[ \|x_i - x_j\|^2 + \eta \|y_i - y_j\|^2 \right] d\pi((x_i, y_i), (x_j, y_j)) \right)^{1/2} \tag{D.2}$$

with $\eta = 10^5$. Both distances are computed globally between samples from the target distribution $q_1$, and samples from $q_0$ simulated forward to $t = 1$ as $\hat{p}_1$.

In addition to the Wasserstein-2 distance, we evaluated the generated conditional distributions using Energy Distance (ED) and Maximum Mean Discrepancy (MMD). Energy Distance is a statistical distance between probability distributions, computed via expected pairwise Euclidean distances between samples:

$$\text{ED}(\hat{p}_1, q_1) = 2\mathbb{E}_{x_i \sim \hat{p}_1, x_j \sim q_1}[\|x_i - x_j\|_2] - \mathbb{E}_{x_i, x_i' \sim \hat{p}_1}[\|x_i - x_i'\|_2] - \mathbb{E}_{x_j, x_j' \sim q_1}[\|x_j - x_j'\|_2] \tag{D.3}$$

We also report the MMD, which quantifies the distance between the mean embeddings of the distributions in a reproducing kernel Hilbert space. To ensure robustness across various spatial scales of the learned distributions, we utilized a multi-bandwidth radial basis function (RBF) kernel, $k(x, x') = \sum_{\sigma \in \mathcal{B}} \exp(-\|x - x'\|_2^2 / 2\sigma^2)$, with bandwidths $\mathcal{B} = \{0.1, 1.0, 10.0\}$. The empirical squared MMD is calculated as:

$$\text{MMD}^2(\hat{p}_1, q_1) = \mathbb{E}_{x_i, x_i' \sim \hat{p}_1}[k(x_i, x_i')] + \mathbb{E}_{x_j, x_j' \sim q_1}[k(x_j, x_j')] - 2\mathbb{E}_{x_i \sim \hat{p}_1, x_j \sim q_1}[k(x_i, x_j)] \tag{D.4}$$

To accurately assess *conditional* generative performance, ED and MMD must be evaluated between distributions sharing equivalent conditions. In cases with discrete conditioning, both ED and MMD were computed on a per-class basis using 2,048 samples and subsequently averaged. For cases with continuous conditioning, we generated 16,384 samples to ensure dense coverage and partitioned the continuous

conditioning variable into 200 equal-width bins. ED and MMD were then computed locally within each bin and averaged across all valid bins to provide a robust measure of conditional statistical fidelity.

### D.4 Material Dynamics Experimental Details

The second case study analyzes the performance of the proposed CVFM algorithm on our target scientific application: modeling the stochastic dynamics of a materials manufacturing process. Unlike the previous examples, the time variable now represents real time in the manufacturing process. In addition, the learned flow passes through a long sequence of empirical intermediate distributions instead of just two boundary distributions.

We use a dataset generated via spinodal decomposition simulations in MEMPHIS (Dingreville et al., 2020), see Appendix C. The dataset contains simulations of the evolution pathway (i.e., trajectory) of 10,000 two-phase microstructures of size $256 \times 256$ pixels. Each simulation is associated with mobility parameters $\mathcal{M}_1$ and $\mathcal{M}_2$, sampled according to the log-uniform $\log(\mathcal{M}) \sim \mathcal{U}(\log(0.1), \log(100))$, and an initial concentration $c_0 \sim \mathcal{U}(-0.7, 0.7)$. These processing parameters correspond to the mobility parameters of the two constituents $(\mathcal{M}_1, \mathcal{M}_2)$, and initial relative concentrations $(c_0)$. Each microstructure's trajectory contains 100 sequential recorded frames representing the microstructure state at a given time (therefore, there are 1M total microstructures spanning 100 time steps). The dataset is split into $8,000$ training trajectories and $2,000$ testing trajectories. The microstructure state at any given time was represented using principal component-based dimensionality reduction of the 2-point statistics, Appendix C. The Principal Component Analysis basis was established based on the complete training dataset. All models were trained to evolve the coefficients of the first 5 PC dimensions.

We note that while complete trajectories for a given $(\mathcal{M}_1, \mathcal{M}_2, c_0)$ are available (and, therefore, paired data is available), the paired data is not utilized in training in order to simulate expected conditions in envisioned experimental applications. The complete trajectories are only used to compute error metrics in testing and to compare against standard methods (such as Neural ODE (Chen et al., 2018) and LSTM (Hochreiter & Schmidhuber, 1997)) that require paired data for training. Instead, state and conditioning pairs are randomly shuffled within each recorded frame in the dataset. They are combined together into joint distributions conditioned on time, $q_{mat}(x, y|\tau)$. Here, $x$ is the PC coefficients representing the microstructure state. $y = (\log(\mathcal{M}_1), \log(\mathcal{M}_2), c_0)$ is the condition vector containing the mobilities and initial concentrations. $\tau$ is the absolute time of the sampled frame. For our example, this empirical distribution contains $800,000$ sample triplets. During training, minor adjustments are made to the underlying CVSFM algorithm (Algorithm 1) to account for the presence of multiple discrete time marginal distributions. Specifically, within each step we first sample a tuple of adjacent time pairs $(\tau_0, \tau_1) \sim \{(\tau_i, \tau_{i+1})\}_{i=0}^{T=99}$. Given this time pair, a mini-batch of size $2B$ $(x, y)$ tuple is drawn for each of these adjacent times. Each time-adjacent mini-batch is then resampled according to the empirical OT coupling (i.e., samples are drawn from the mini-batch estimate of the optimal coupling). Lastly, the standard CVFM algorithm is adjusted such that $t \sim \mathcal{U}(0, 1)$ and then linearly mapped to $\tau = \tau_0 + t(\tau_1 - \tau_0)$ where $\tau \in [\tau_0, \tau_1]$. This complete CVSFM algorithm for stochastic dynamics is outlined in Algorithm 2.

Mean absolute error values, along with minimum and maximum absolute error reported in Table 2 were evaluated on a test set of 2,000 microstructures, and a training set of 8,000 microstructures. Known state was compared against the expectation of the predicted conditional distribution over the state variable at a given time $\mathbb{E}[x_t|x_0, y]$.

All material specific cases used networks with five layers of width 256 and GELU activations (Hendrycks & Gimpel, 2023). Skip connections were applied over the middle three layers. The conditioning variable was subject to both self-attention and time cross-attention with an embedding dimensionality of 64 and 8 heads. The same network architecture was used in T-COT-FM, and for both drift and score networks in CVSFM, COT-SFM, and T-COT-SFM. It was duplicated for the Neural ODE benchmark. The LSTM benchmark consisted of four layers with width 512 for approximately equivalent parameterization. Optimization was carried out with cosine annealing of the learning rate from $1e-3$ to $1e-8$ and ADAM-W (Loshchilov & Hutter, 2019) with weight decay $1e-2$ and an effective batch size of 1024. Conditional probability paths were constructed with $\sigma_x = 0.1$, $\sigma_y = 0.1$, and mini-batch conditional OT was performed with $\eta = 100$.

---

**Algorithm 2** Time-Specific Conditional Variable Score and Flow Matching for Stochastic Dynamics

---

**Require:** Source and target data distributions $q(x_0, y_0)$ and $q(x_1, y_1)$ consisting of two unpaired sets of observed samples, noise hyperparameters $\sigma_x$, $\sigma_y$, mini-batch size $B$, weighting schedule $\lambda(t)$, drift network $v_\theta$, and score network $s_\theta$.

 **while** Training **do**
  $(\tau_0, \tau_1) \sim \{(\tau_i, \tau_{i+1})\}_{i=0}^{T=99}$           $\triangleright$ Randomly select adjacent times.
  $\{(x_0, y_0), (x_1, y_1)\}_{b=1}^{B} \sim q(x_0, y_0|\tau_0), q(x_1, y_1|\tau_1)$
  $\pi \leftarrow \text{OT}(\{(x_0, y_0), (x_1, y_1)\}_{b=1}^{B})$    $\triangleright$ Interchangeable with Sinkhorn algorithm (Cuturi, 2013).
  $\{(x_0^{(b)}, x_1^{(b)}), (y_0^{(b)}, y_1^{(b)})\}_{b=1}^{B} \sim \pi(z, w)$     $\triangleright$ Resample from time-specific OT coupling.
  $t \sim \mathcal{U}(0, 1)$
  $\tau \leftarrow \tau_0 + t(\tau_1 - \tau_0)$
  $z = \left[(x_0^{(1)}, x_1^{(1)}), \ldots, (x_0^{(B)}, x_1^{(B)})\right], \quad w = \left[(y_0^{(1)}, y_1^{(1)}), \ldots, (y_0^{(B)}, y_1^{(B)})\right]$
  $\alpha(w) \leftarrow \exp(-\|y_0 - y_1\|_2^2 / 2\sigma_y^2)$      $\triangleright$ Evaluate kernel for sampled conditioning values.
  $p_t(x|z) \leftarrow \mathcal{N}(x; tx_1 + (1-t)x_0, \sigma_x^2 t(1-t))$
  $p_t(y|w) \leftarrow \mathcal{N}(y; ty_1 + (1-t)y_0, \sigma_y^2 t(1-t))$
  $x_t \sim p_t(x|z)$
  $y_t \sim p_t(y|w)$
  $\hat{u}_t(x|z) \leftarrow ((1-2t)/t(1-t))(x - (tx_1 + (1-t)x_0)) + (x_1 - x_0)$
  $\nabla_x \log p_t(x|z) \leftarrow (tx_1 + (1-t)x_0 - x)/(\sigma_x^2 t(1-t))$
  $\mathcal{L}_{\text{CVSFM}} \leftarrow \mathbb{E}_{t, q(z,w), p_t(x,y|z,w)}\left[\alpha(w)\left(\|v_\theta(x, y, \tau) - \hat{u}_t(x|z)\|^2 + \lambda(t)^2\|s_\theta(x, y, \tau) - \nabla_x \log p_t(x|z)\|^2\right)\right]$
  $\theta \leftarrow \text{Update}(\theta, \nabla_\theta \mathcal{L}_{\text{CVSFM}})$
 **end while**
 **return** $v_\theta, s_\theta$

---

While, T-COT-FM in its initial formulation does not include a varying noise schedule across the conditioning variable, we have also included an ablation with this extension, mirroring the same value of $\sigma_y$.

## D.5 MNIST-FashionMNIST Experimental Details

In addition to the presented case studies in the main body, we pursued additional experiments to understand CVFM's performance in other situations. For example, we studied image-based domain transfer to begin to analyze the performance of CVFM on high-dimensional problems. In this setting, we explore conditional domain transfer; we map MNIST digits to classes in the FashionMNIST dataset, Figure E.7. Specifically, the source distribution is the MNIST dataset where $x_0$ are images and $y_0$ are digit assignments. Similarly, $x_1$ and $y_1$ are the images and class assignments in the FashionMNIST dataset. See Appendix E.3 for the results for this case study. In training, we unpair the datasets and draw samples from each marginal randomly. We emphasize that because this is a discrete condition setting, the probability of getting true pairs within a batch remains high. This makes the problem *significantly easier* than with continuous conditioning. Therefore, this case study simply tests whether the additions made in CVFM can help minimize the impact of the unpaired marginals and stabilize the simple high-dimensional problem.

In particular, the study analyzes whether training remains stable when using the scaling kernel, $\alpha(w)$, on high-dimensional problems (see Appendix A for theoretical analysis of the scaling kernel $\alpha(w)$). In fact, we see that training stability improves slightly – mirroring the conclusions from the toy examples – even for the high-dimensional setting.

The MNIST-FashionMNIST domain transfer experiments utilized a UNet architecture developed by OpenAI (`https://github.com/openai/guided-diffusion/tree/main`). The network configuration utilized in this work consisted of 64 channels, with channel multiples of [1,2,2,2]. 4 heads of self-attention over 16 and 8 resolution were applied with 2 residual blocks. Optimization was carried out with cosine annealing of the learning rate from $1e-4$ to $1e-8$ and ADAM-W (Loshchilov & Hutter, 2019) with weight decay $1e-4$ for ranges of 3,750 - 10,000 epochs and batch size of 1024, or equivalently 220,000 - 590,000

steps. Conditional probability paths were constructed with $\sigma_x = 0.1$, $\sigma_y = 1e-3$, and mini-batch conditional OT was performed with $\eta = 10$ and $\eta = 1000$.

Unconditional FID scores (i.e., FID scores computed over the marginal distribution on $x_1$ to compare the target ground truth and the predicted target) were computed over 10,000 samples using *Clean-FID* (`https://github.com/GaParmar/clean-fid`). Conditional FID scores were computed using the same algorithm, however the calculation was restricted to a subset of the total samples with matching conditioning (1,000 samples per class). This methodology directly captures the distributional match in a conditional sense, with ground truth target samples corresponding to a given condition being compared against samples from the source distribution with the same condition that had been evolved through time using the learned network. LPIPS scores were computed using *torchmetrics* (`https://github.com/Lightning-AI/torchmetrics`).

# E    Additional Case Study Specific Results

We move on towards presenting additional results obtained throughout this work. The results expand upon the discussion presented in Section 5 towards interrogating the empirical performance of CVFM, illuminating the critical advances necessary for learning conditional vector fields. We refer the reader to Appendix D for detailed descriptions of the datasets and training infrastructure used in each case study.

## E.1    2D Experiments

In Figure E.1, trajectories of the learned vector fields were presented for the *8 Gaussians – Moons* mapping with discrete conditioning and the *Moons – Moons* with continuous conditioning, although this remains a fraction of the cases run. For completeness, we present the trajectories of all methodologies evaluated in Table 1 in Figure E.1.

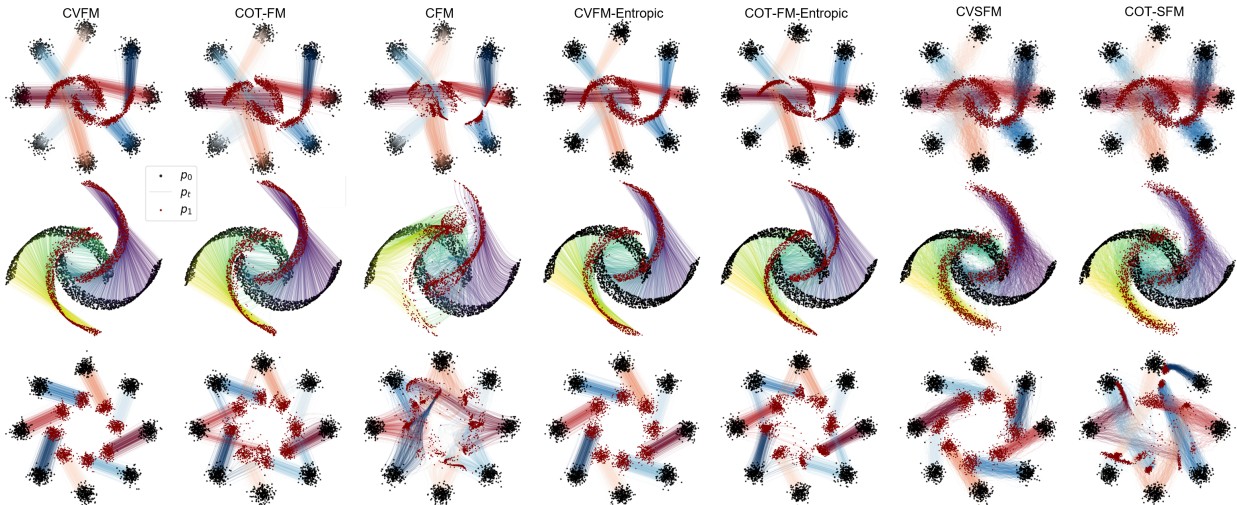

Figure E.1. Comparison of obtained trajectories for various OT and SB modeling approaches in the synthetic datasets considered associated with Wasserstein-2, Energy Distance, and Maximum Mean Discrepancy error reported in Table 1. Trajectories are colored by the conditioning variable. Source distributions are shown in black, target distributions are shown in maroon.

While, perhaps visually the simplest, the *8 Gaussians – 8 Gaussians* mapping, consisting of a 45 degree rotation about the origin, is demonstrably the most complex family of conditional vector fields to learn given samples of the empirical distribution $q(w, z)$. This is the only case in which only CVFM variants reliably learn to disentangle the conditional dynamics (Figure E.1), while COT-FM attempts to split mass

to minimize transport costs across the joint $\mathcal{X} \times \mathcal{Y} : \mathbb{R}^N \times \mathbb{R}^M$, visualized by splits mapping to target densities with similar conditioning values.

**Conditional Transport Cost**: This mass splitting observed in the *8 Gaussians – 8 Gaussians* with COT-FM, alongside observed biasing effects in the *8 Gaussians – Moons* and *Moons – Moons* cases is a direct result of conditional transport cost defined in Eq. (9). The role of $\eta$ in penalizing transport in the conditioning variable specifically dictates performance of the learned mappings of both COT-FM and CVFM, although it has a more pronounced impact on the former. Additional ablations across all 2D cases can be seen in Figure E.2. Across all cases, a Wasserstein-2 error gap can be observed with the exception of *8 Gaussians – Moons* with $\eta > 25$. This only further reinforces earlier insights, highlighting the improved robustness of CVFM.

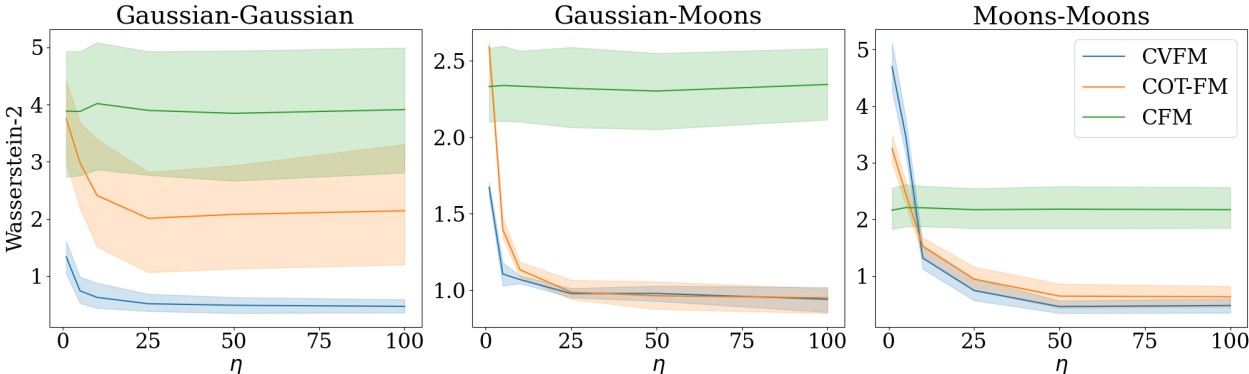

Figure E.2. Parameter study analyzing the sensitivity of the final model prediction to the choice of the $\eta$-hyperparameter. Recall, $\eta$ is the weight on the conditional term in the OT ground cost. The study compares three models: the proposed CVFM, COT-FM (roughly, CVFM without the kernel), and CFM (CVFM without the OT or the kernel). Due to the contribution from the incorporated kernel, CVFM is only minimally sensitive to the choice of $\eta$.

### E.2 Supplementary Analysis of Estimator Variance

**Conditioning Mismatch Kernel**: The conditioning mismatch kernel $\alpha(w)$ plays a pivotal role in the objective introduced in Eq. (7), which we duplicate here in its complete form.

$$\mathcal{L}_{\text{CVFM}}(\theta) = \mathbb{E}_{t,q(z,w),p_t(x|z)p_t(y|w)} \left[ \alpha(w) \| v_\theta(x, y, t) - u_t(x|z) \|^2 \right] \tag{E.1}$$

The selected form of this kernel dictates the degree of continuity expected *a priori* in the observed joint vector fields $u_t(x, y)$ across $y \in \mathbb{R}^M$, introducing an inductive bias in the solution across this joint space, even if we only ever expect to evaluate the vector field in a conditional sense. In this work, we select the Squared Exponential (SE) kernel $\alpha(w) = \exp(-\|w\|_2^2/2\sigma_y^2)$ with observation $w = y_1 - y_0$, where the degree of continuity in $u_t$ across the conditioning variable can be tailored through an appropriate selection of $\sigma_y$.

Previously, in Table 1, we observed significantly improved performance with the introduction of this additional modulation – which naturally raises questions as to its contribution in isolation. Figure E.4 demonstrates that even by itself in the absence of mini-batch OT, the introduction of $\alpha(w)$ is able to reliably equal or improve upon Wasserstein-2 target distribution error in comparison with COT-FM and CVFM. Figure E.5 similarly shows an evaluation across $\eta$ values (the weighting in the optimal transport ground cost), highlighting the stability it introduces across all test cases relative to mini-batch OT solves with Eq. (9). In the *8 Gaussians – 8 Gaussians* case, $\alpha(w)$ drastically outperforms COT-FM, with comparable performance in the other mappings, albeit without providing approximate OT within the conditioned vector fields. Due to this limitation, one might view $\alpha(w)$ in isolation as a reliable extension to conditional CFM.

**Objective Target Variance**: One reason for the improved convergence of CVFM is the significantly lower variance of the training objective target in CVFM in comparison to similar existing methods such as COT-

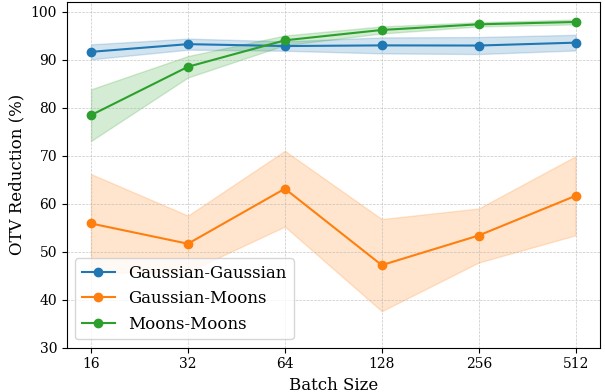

Figure E.3. Percentage reduction in variance (measured as $\mathbb{E}_{t,q,p_t}\|u_t(x|z)\|^2$) achieved by CVFM relative to COT-FM. Across all batch sizes, the inclusion of the kernel $\alpha(w)$ consistently eliminates a significant portion of the estimator variance. Notably, in the *8 Gaussians – 8 Gaussians* topology, which features the strongest conditional dependence, CVFM reduces variance by over 90% regardless of the sampling sparsity.

|  | Batch Size | 8 Gaussians – 8 Gaussians | 8 Gaussians – Moons | Moons – Moons |
|---|---|---|---|---|
| CVFM | 16 | 6.465±0.561 | 7.847±1.405 | 0.699±0.157 |
|  | 32 | 2.896±0.364 | 3.618±0.142 | 0.179±0.030 |
|  | 64 | 1.515±0.200 | 1.648±0.268 | 0.044±0.007 |
|  | 128 | 0.678±0.052 | 0.946±0.099 | 0.014±0.002 |
|  | 256 | 0.379±0.086 | 0.493±0.057 | 0.005±0.001 |
|  | 512 | 0.175±0.032 | 0.216±0.043 | 0.002±0.000 |
| COT-FM | 16 | 77.376±12.805 | 17.780±2.657 | 3.243±0.356 |
|  | 32 | 42.891±4.948 | 7.482±0.857 | 1.561±0.154 |
|  | 64 | 21.197±0.802 | 4.468±0.621 | 0.739±0.043 |
|  | 128 | 9.672±2.178 | 1.791±0.266 | 0.369±0.049 |
|  | 256 | 5.383±0.588 | 1.057±0.037 | 0.191±0.026 |
|  | 512 | 2.718±0.475 | 0.563±0.045 | 0.094±0.012 |

Table E.1. Variance of the conditional objective across synthetic datasets and varying batch sizes. Variance values were computed over 5 seeds with a conditioning transport weight of $\eta = 10$. Values are reported as $\mu \pm \sigma$.

FM Kerrigan et al. (2024); Chemseddine et al. (2024). As discussed in Section 3.3, we measure this variance using the objective target variance (OTV):

$$\text{OTV} = \mathbb{E}_{t,q(z,w),p_t(x,y|z,w)}\|u_t(x|z)\|^2 \tag{E.2}$$

Notably, the absolute computed OTV values generally should not be compared across different test case studies because these equations are not normalized to the scale of the true underlying conditional vector field. Figure E.3 visualizes the relative reduction in variance as a function of batch size. As discussed in Section 3.3, the kernel leads to consistent reductions in variance across all applications. In practice, this characteristic is immensely valuable because it combats the well known instability of flow matching training, effectively reducing the number of necessary training steps (e.g., see Figure E.1). In Figure E.3 we see that this reduction is remarkably consistent across batch sizes (e.g., the kernel achieves a reduction of roughly 90% for the challenging *Gaussian – Gaussian* topology). This indicates that the kernel continues to provide benefits even when other variance reduction methods are used as well. Table E.1 details the raw variance values, highlighting that while the *percentage* improvement is constant, the *absolute* stability gain is most critical in sparse regimes (e.g., reducing target variance from $77.38 \rightarrow 6.47$ at $B = 16$).

**Kernel Sensitivity Ablation**: While the results above demonstrate the utility of the SE kernel, a natural question arises regarding the sensitivity of CVFM to the specific functional form of $\alpha(w)$. To address this, we evaluated three kernels with distinct tail behaviors to test the robustness of the variance reduction mechanism:

$$\text{Squared Exponential (light-tailed)}: \quad \alpha_{\text{SE}}(w) = \exp\left(-\frac{\|w\|_2^2}{2\sigma^2}\right),$$

$$\text{Laplace (moderate tails)}: \quad \alpha_{\text{Lap}}(w) = \exp\left(-\frac{\|w\|_1}{\sigma}\right),$$

$$\text{Cauchy (heavy-tailed)}: \quad \alpha_{\text{Cauchy}}(w) = \left(1 + \frac{\|w\|_2^2}{\sigma^2}\right)^{-1}.$$

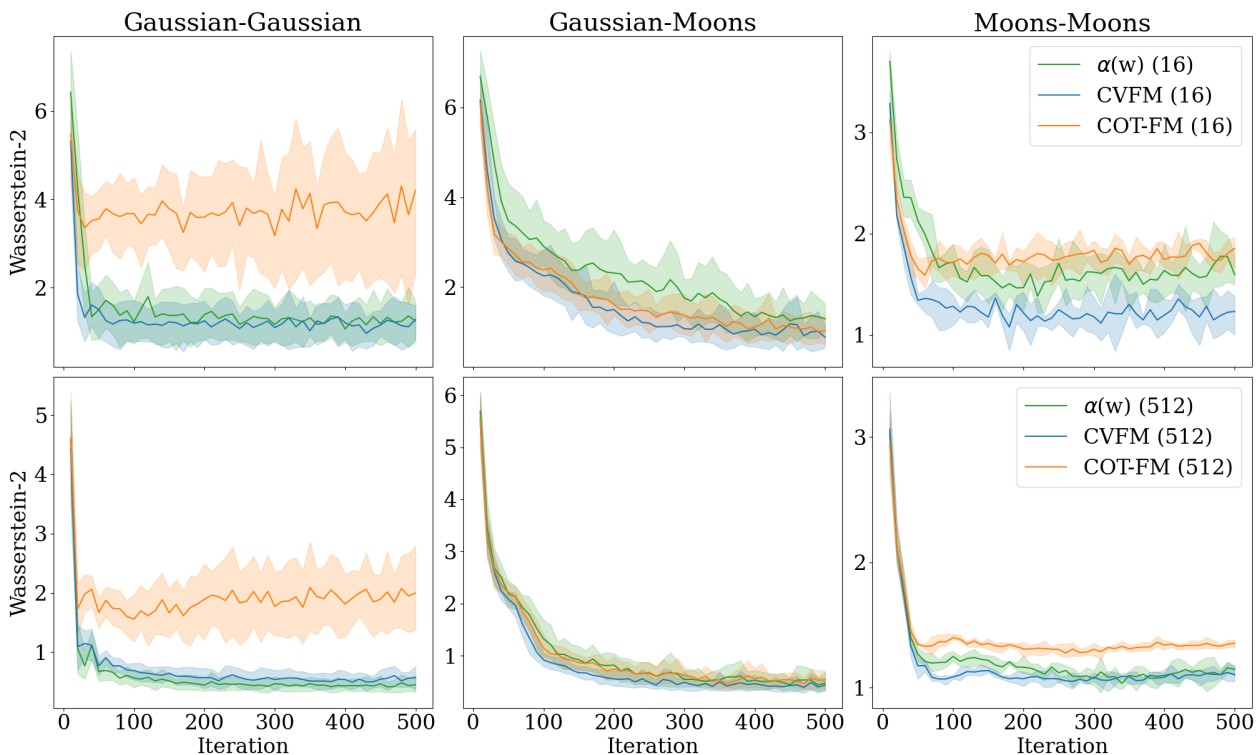

Figure E.4. Demonstration of the effectiveness of performing an expectation of the objective with solely $\alpha(w)$ in comparison to the COT-FM and CVFM approaches. The use of $\alpha(w)$ in isolation is better able to disentangle associated conditioning variables in almost all cases than the conditional Wasserstein distance introduced in Eq. (9).

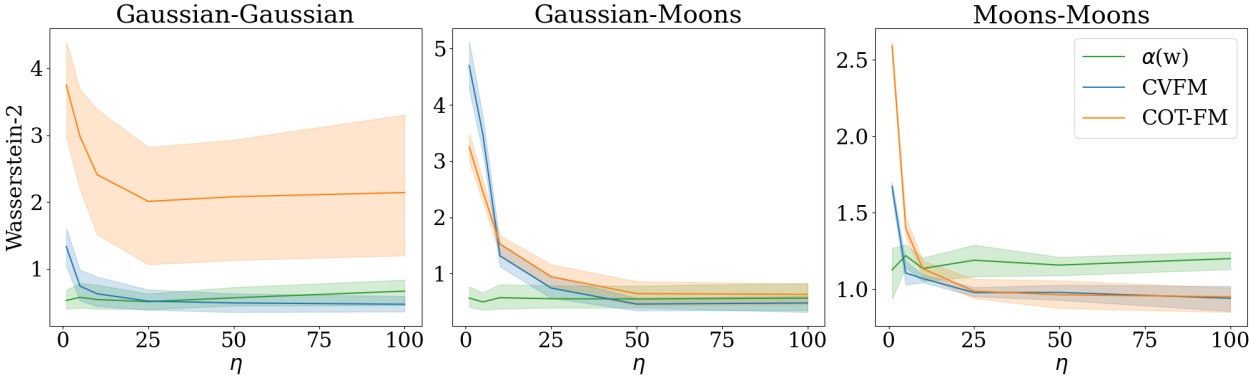

Figure E.5. $\alpha(w)$ more reliably reduces Wasserstein-2 error to the target distribution in comparison with COT-FM across values of $\eta$.

As shown in the left portion of Figure E.6, the choice of kernel has a measurable but minor impact on estimator stability compared to the baseline. The ordering of variance corresponds to the kernel's decay for a fixed bandwidth. The Squared Exponential (SE) kernel achieves the lowest variance due to its rapid decay and strict locality, suppressing noisy, distant pairings. The Cauchy kernel, with its heavy tails, allows more distant conditioning pairs to contribute to the gradient; while this increases the effective sample size, it introduces slightly more variance. However, the tight clustering of all three variants confirms that the method is robust: provided that *any* monotonically decaying kernel is used to enforce local consistency, the catastrophic variance explosion of COT-FM displayed in Figure 2 is avoided. The right portion of Figure

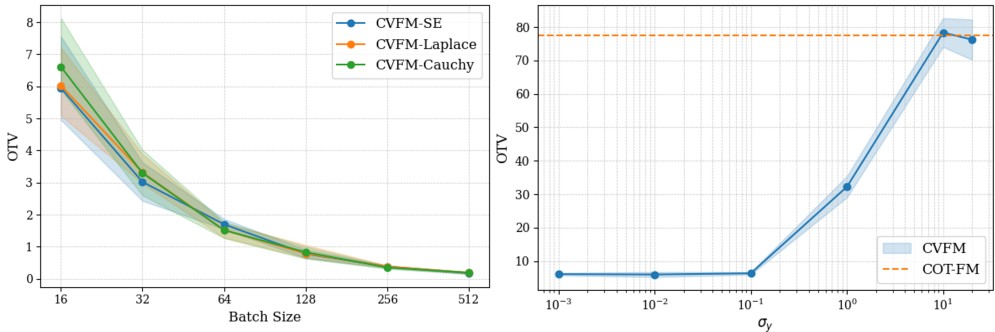

Figure E.6. Comparison of the three ablated kernels (left), and ablation of kernel bandwidth for the SE kernel (right) on the *8 Gaussians – 8 Gaussians* topology with constant batch size ($B = 16$) and conditional ground cost ($\eta = 10$), highlighting the robust variance reduction across varied bandwidths.

E.6 demonstrates the impact of the SE's kernel bandwidth hyperparameter on target variance – highlighting the exceptional stability across a wide range spanning orders of magnitude relative to the COT-FM baseline.

Beyond empirical performance, the squared-exponential (SE) kernel is a natural choice from the perspective of entropic optimal transport (EOT). In EOT, the entropic regularization of the Kantorovich problem yields transport plans of Gibbs form

$$\pi_\varepsilon \propto \exp\left(-\frac{C(y_0, y_1)}{\varepsilon}\right). \tag{E.3}$$

When the ground cost is the squared Euclidean distance $C(y_0, y_1) = \|y_0 - y_1\|_2^2$, the Gibbs factor is Gaussian in the vector difference $y_0 - y_1$, and thus an SE kernel $\alpha(y_0, y_1) \propto \exp(-\|y_0 - y_1\|^2/(2\sigma^2))$ arises (with $\sigma^2 = \varepsilon/2$, up to normalization). This provides a principled link between SE weighting and a soft, entropically-regularized preference for local conditioning. By contrast, heavy-tailed or non-smooth kernels (for example Cauchy or Laplace) do not correspond to the Gibbs factor for the squared-$L_2$ cost; (Laplace kernels do correspond to an $L_1$ cost, while Cauchy kernels cannot be written as a simple Gibbs factor of a quadratic cost). This observation plausibly explains the weaker variance reduction we observed for those kernels in our ablation. We therefore present the SE kernel as a theoretically motivated default, while still reporting improvements over COT-FM when other kernels are used.

### E.3 MNIST-FashionMNIST Domain Transfer

In this section, we evaluate our method in high-dimensional domain transfer between MNIST digits and FashionMNIST clothing articles with conditioning by discrete class, illustrating the methodology's performance in higher-dimensional distributional mappings. We directly compare our two variants, CVSFM against COT-SFM, Figure E.7. The FID and LPIPS scores reported are averages across the class conditioned scores (i.e., FID and LPIPS are computed on a per-class basis across 1,000 samples in each class). Select images from the source and corresponding generated images from the target conditional distributions are also shown. As seen in the repeated samples, the model is able to consistently map to the correct paired conditional distribution while displaying appreciable diversity within each class. Mirroring the 2D experiments, we observe improved convergence and mode coverage with CVSFM compared to COT-SFM across all $\eta$ values due to $\alpha(w)$ modulating conditional flow.

This improved convergence with increasing $\eta$ of COT-FM only further reinforces the value of ground cost scaling across $y$. Unfortunately, the optimal value of $\eta$ is problem dependent, and is particularly challenging to identify a priori. The conditional scaling kernel ameliorates these difficulties. As shown in Figure E.3, the kernel significantly reduces the objective variance in comparison with purely conditional OT, facilitating stable convergence and more accurate conditional mappings. This behavior was first observed in 2D examples in Figure E.1 and such characteristics extend to the high-dimensional setting. This increased stability facilitates a greater tolerance on $\eta$ values, ameliorating potential difficulties during hyperparameter optimization. In Figure E.8, we observe this stability through the inspection of randomly generated samples

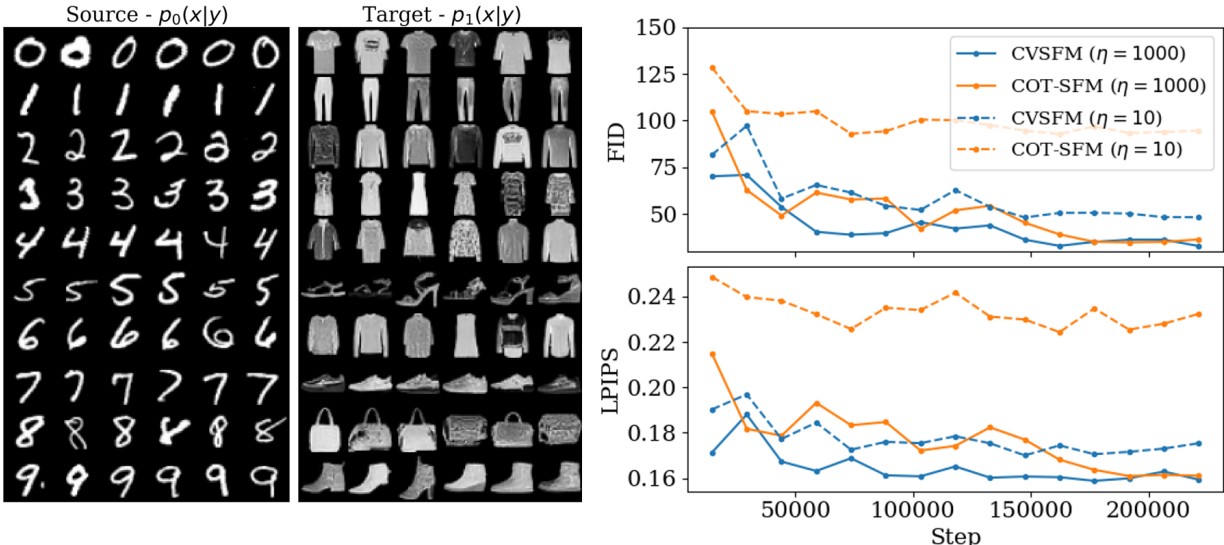

Figure E.7. CVSFM conditionally generated images from the FashionMNIST dataset. (Left) pane displays samples from the initial distributions $p_0(x|y)$, while the middle displays generated samples from $p_1(x|y)$. Relative positioning indicates paired samples. (Right) pane illustrates the improved convergence of CVSFM over COT-SFM in high-dimensional domain transfer for $\eta = 10$ and $\eta = 1000$. Displayed FID/LPIPS scores are computed per class and averaged.

Table E.2. Comparison of conditional mean (FID) and unconditional FID scores (FID-All) for 10,000 samples alongside conditional LPIPS scores.

| Model | FID-All ($\downarrow$) | FID ($\downarrow$) | LPIPS ($\downarrow$) |
|---|---|---|---|
| CVSFM ($\eta = 1000$) | 11.668 | 32.915 | 0.159 |
| COT-SFM ($\eta = 1000$) | 14.965 | 36.456 | 0.161 |
| CVSFM ($\eta = 10$) | 23.554 | 48.326 | 0.175 |
| COT-SFM ($\eta = 10$) | 15.751 | 94.690 | 0.232 |

from mapping the first class of MNIST to the first class of FashionMNIST. While in an unconditional sense, COT-SFM with $\eta = 10$ is able to appropriately transfer between MNIST and FashionMNIST, its consistency in mapping to the correct conditional distribution is lost without elevated penalization in transport across $y$. In comparison, CVSFM is able to consistently map to t-shirts/tops in the first class even with orders of magnitude difference in $\eta$. Similarly, in Figure E.7, the convergence behavior for CVFM degrades very little with large changes in $\eta$.

Figure E.7 displayed the convergence characteristics of conditional FID and LPIPS scores, conditionally evaluated for each class of $p_1(x|y)$, only serving to reinforce the prior discussion. In comparison, in evaluating FID scores for $p_1(x) = \int p_1(x|y)p(y)dy$, distinctions in the performance between CVSFM and COT-SFM are removed. Figure E.9 highlights the equivalence in unconditional performance in this case study across 10,000 images. This discrepancy in conditional to unconditional FID scores highlights limitations of mini-batch sampling from the conditional OT coupling $\pi_\eta((x_0, y_0), (x_1, y_1))$. Even with elevated weighting on transport in the conditioning variable, mini-batch conditional OT provides a poorer approximation.

## E.4 Material Dynamics

The mean absolute error metrics displayed in Table 2 provide point estimates of the performance of our proposed method, providing evidence that we are capable of reliably disentangling the latent processing dynamics of material microstructures across the conditional processing space, given unpaired samples of

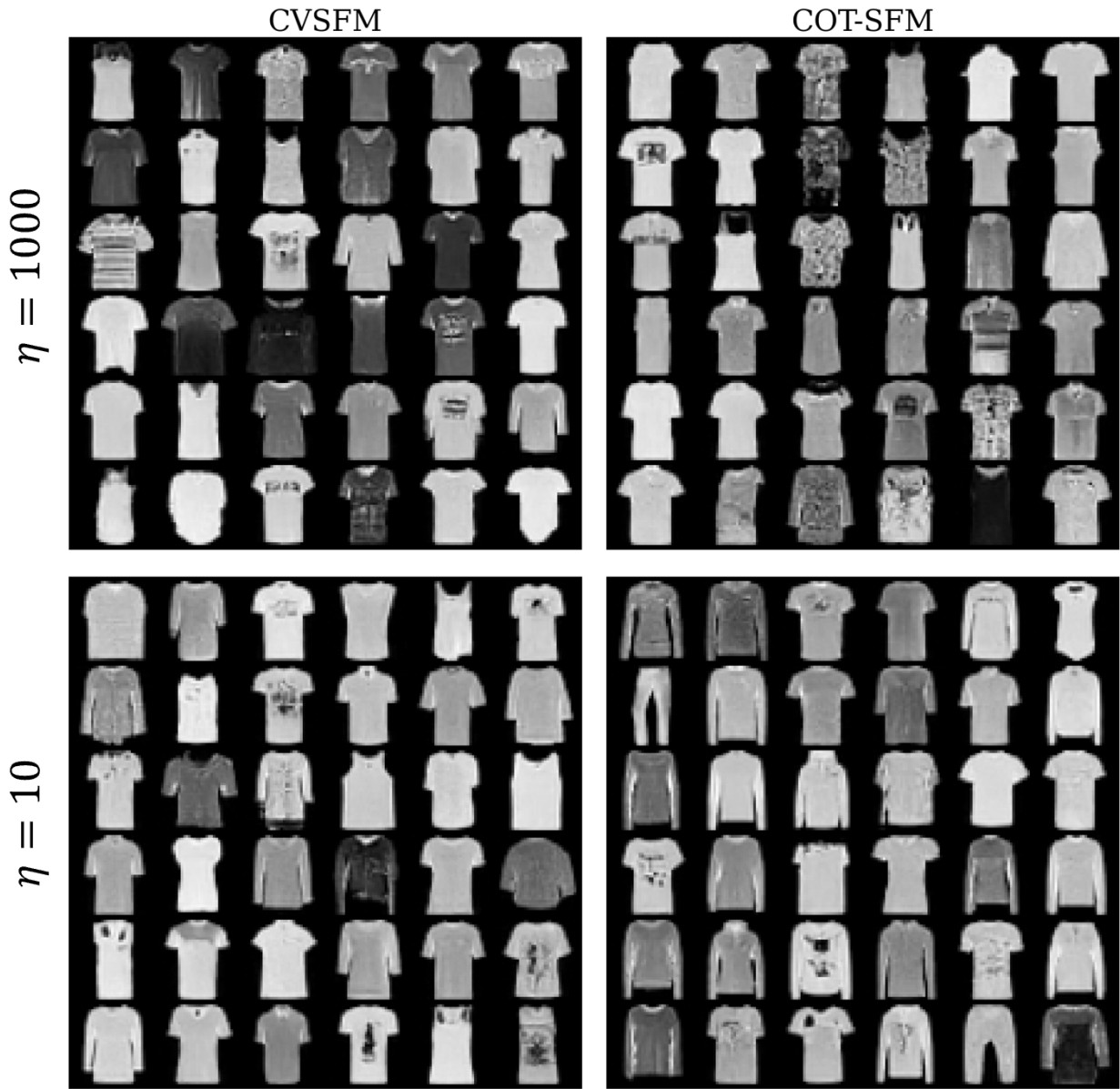

Figure E.8. Comparison of 32 randomly generated images corresponding to the first class of FashionMNIST with CVSFM and COT-SFM for $\eta = 10$ and $\eta = 1000$.

process conditions and material state observed at discrete times. For further interrogation, we present the estimated error distributions in Figure E.10 across all 2,000 material samples in the test set. In congruence with the results presented earlier, CVSFM-Exact outperforms all other methods, a surprising fact given the stochastic nature of only viewing observations sampled from $q(z, w, \tau)$. The Neural ODE (Chen et al., 2018) outperforms other methods, approximately matching the performance of the best CVSFM-Exact variant (i.e., exhibiting similar tails and a mean shift). Figure E.11 compares the dynamics predicted by the three models on several selected members of the test dataset. Mirroring the distribution of errors in Figure E.10, the CVSFM model slightly outperforms the other two while also providing uncertainty estimates. The panels in Figure E.11 display the dynamics projected onto individual principal component subspaces, $\alpha_i$.

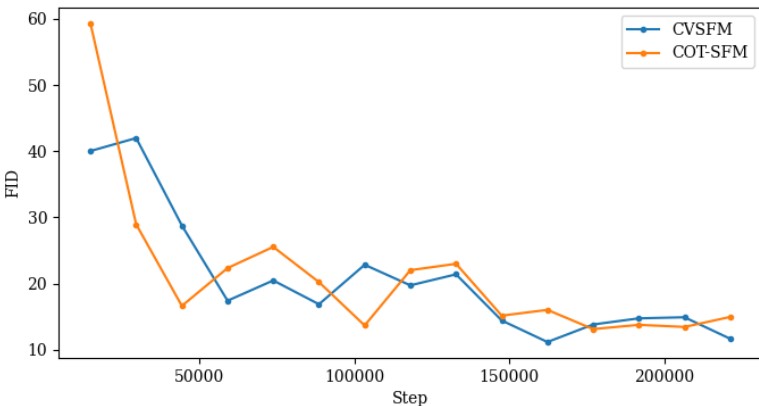

Figure E.9. Unconditional FID scores evaluated across 10,000 samples $x \sim p_1(x)$ for COT-SFM and CVSFM ($\eta = 1000$) during training.

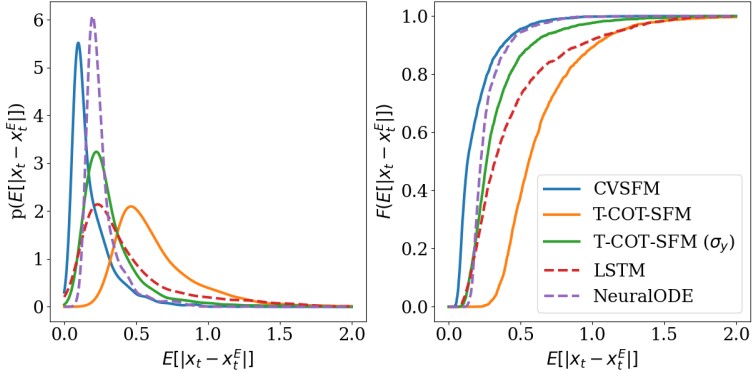

Figure E.10. (Left) probability density function, and (right) cumulative distribution function of CVSFM in comparison with evaluated conventional approaches requiring complete trajectory information. Errors are computed between entire trajectories – here, model predictions are made by rolled-out predictions initialized at $t = 0$.

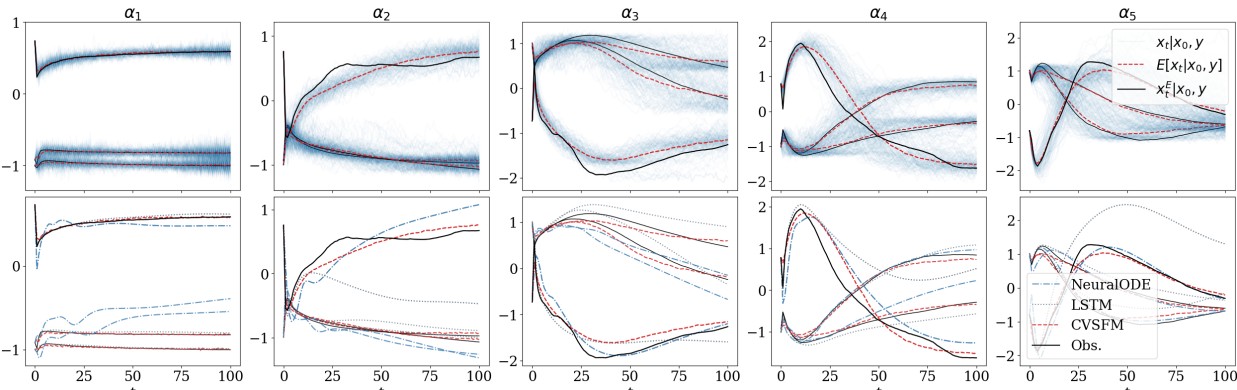

Figure E.11. Collection of three randomly sampled trajectories from the test set in PC space displaying (top) 128 samples in blue from CVSFM, with the expected value in red, and (bottom) deterministic predictions of the LSTM and Neural ODE shown against the expected value of CVSFM.

## F  Hyperparameter Guidance

The proposed CVFM framework introduces two primary hyperparameters governing the conditional coupling: the kernel bandwidth, $\sigma_y$, and the transport penalty, $\eta$. While our ablations (Figures E.1, E.5, E.6) demonstrate that CVFM is robust to hyperparameter choice compared to unregularized baselines, we offer the following practical heuristics for their selection in new applications.

**Kernel Bandwidth ($\sigma_y$):** The bandwidth $\sigma_y$ in the conditioning mismatch kernel, $\alpha(w) = \exp(-\|y_0 - y_1\|_2^2/2\sigma_y^2)$, defines the effective scale of local support over which conditional vector fields are aggregated.

- *Heuristic*: We recommend setting $\sigma_y$ to approximately the **10th percentile** of Euclidean distances between randomly sampled conditioning pairs. This tighter bandwidth ensures the kernel focuses on the local support of the conditioning manifold, effectively filtering out disparate conditions while remaining robust to outliers.

- *Sensitivity*: As illustrated in Figure E.6 (Right), the method exhibits stability across bandwidths spanning several orders of magnitude ($10^{-1}$ to $10^1$). Extremely small values of $\sigma_y$ approach the regime of exact matching (reducing effective sample size), while excessively large values approach the unregularized limit. The broad plateau of low objective target variance suggests that precise tuning is rarely necessary, provided $\sigma_y$ restricts interaction to the local scale of the conditioning data.

**Transport Penalty ($\eta$):** The penalty $\eta$ in the anisotropic ground cost (Eq. 9) biases the Optimal Transport solver to prefer couplings with matched conditioning variables ($y_0 \approx y_1$).

- *Heuristic*: We recommend setting $\eta$ sufficiently high (e.g., $\eta \in [10, 100]$) to strongly penalize transport along the conditioning manifold, effectively acting as a soft constraint.

- *Interplay with CVFM*: In standard Conditional OT (COT-FM), performance is highly sensitive to $\eta$, as the solver must rigidly enforce conditional constraints to avoid mismatched pairings. However, CVFM relaxes this dependency. As shown in Figure E.5, the conditioning mismatch kernel $\alpha(w)$ effectively filters out high-cost pairings from the OT solve, maintaining low Wasserstein-2 error even when $\eta$ is suboptimal. Consequently, users should prioritize a large enough $\eta$ to guide the coarse coupling, relying on the kernel to handle local consistency.

## G  Computational Cost and Scalability

To evaluate the computational overhead and scalability of CVFM, we analyzed the training runtime and corresponding performance across our 2D synthetic benchmarks. Specifically, we isolated the impact of the optimal transport (OT) solver by comparing three coupling strategies: Exact OT, entropically-regularized OT (Sinkhorn), and the use of the conditioning mismatch kernel $\alpha(w)$ in isolation without an explicit OT solver.

We analyzed the training runtime, memory footprint, and corresponding performance across our 2D synthetic benchmarks. Theoretically, the primary computational bottleneck of the exact optimal transport (OT) solver is its time complexity, which scales cubically, typically $\mathcal{O}(B^3 \log B)$, with batch size $B$. In contrast, peak GPU memory utilization remains largely bottlenecked by the network's forward and backward passes; the $\mathcal{O}(B^2)$ memory footprint required to store the pairwise distance matrix for the OT solve is negligible for typical batch sizes (e.g., $B \leq 512$). Furthermore, while the OT solver itself is independent of data dimensionality, the preliminary step of computing the pairwise cost matrix scales strictly linearly $\mathcal{O}(D)$. Therefore, moving to higher-dimensional conditional spaces impacts the runtime linearly, reinforcing that batch size–rather than dimensionality or memory–is the true limiting factor for exact mini-batch OT.

This theoretical bottleneck is empirically reflected in the top row of Figure G.1, which illustrates the wall-clock training time as a function of batch size. As expected, the Exact OT solver exhibits poor computational scalability, with training times growing steeply at larger batch sizes. While the Sinkhorn algorithm provides

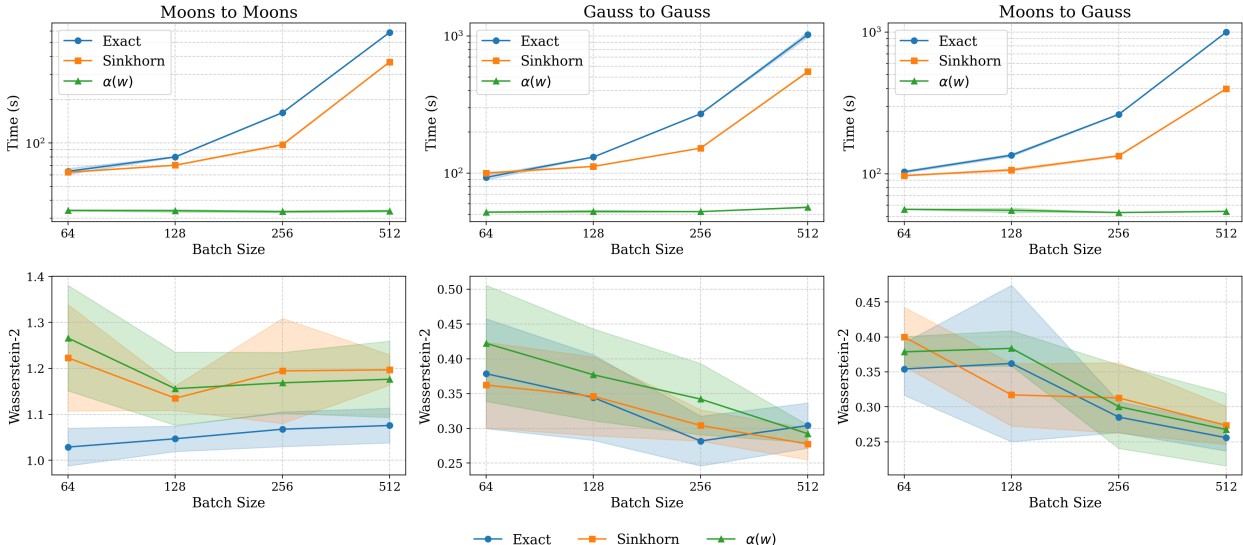

Figure G.1. Computational cost comparison for a single batch of CVFM implemented using three Optimal Transport Strategies: Exact (calculated via Exact OT), Sinkhorn (Entropically Regularized OT calculated via Sinkhorn's algorithm), and $\alpha(w)$ (just the kernel is used without any other OT). The first row compares the time required to perform calculations on one batch as a function of the batch size for the three toy problems considered previously. Row 2 compares the final model performance achieved by the model settings from Row 1, measured via the Wasserstein-2 between the model predictions and the ground truth.

a faster, entropically-regularized alternative, its computational overhead remains substantial as batch sizes scale. In stark contrast, relying solely on the $\alpha(w)$ kernel bypasses the mini-batch assignment problem entirely. This reduces the operation to a highly efficient metric evaluation, circumventing the $\mathcal{O}(B^3 \log B)$ bottleneck and yielding a nearly flat, highly scalable runtime profile that requires only a fraction of the time compared to the OT solvers at $B = 512$.

A critical consideration is whether omitting the OT solver and relying exclusively on the variance-reducing properties of $\alpha(w)$ degrades the model's ability to learn conditional dynamics. The bottom row of Figure G.1 demonstrates that $\alpha(w)$ in isolation remains remarkably competitive. In the *8 Gaussians – 8 Gaussians* and *Moons – 8 Gaussians* topologies, the isolated kernel outperforms Exact OT and achieves a Wasserstein-2 error that closely matches the performance of the Sinkhorn algorithm. In the continuous *Moons – Moons* scenario, the kernel maintains comparable performance to both OT baselines, albeit with slightly higher variance at smaller batch sizes.

These benchmarks highlight a highly practical efficiency trade-off for practitioners. While solving mini-batch OT (particularly via Sinkhorn) provides robust, explicitly optimized conditional pairings, it introduces a significant computational bottleneck. The conditioning mismatch kernel $\alpha(w)$ not only acts as a critical variance-reduction mechanism for standard CVFM, but can also serve as an extremely lightweight standalone alternative. In computationally constrained regimes, or when scaling to very large batch sizes and high-dimensional conditioning spaces, utilizing $\alpha(w)$ in isolation provides a sufficient strategy to enforce local conditional consistency without the prohibitive cost of optimal transport solves.

