# OpenReview forum: "Modeling Stochastic Conditional Dynamics from Sparse Observations via Kernel-Stabilized Flow Matching"
_TMLR — Accepted by TMLR_

### Review · Reviewer_mW6b · 2026-02-20

**Summary Of Contributions:**

This paper addresses the challenge of modelling the evolution of time-dependent conditional probability densities, using sparse and unpaired observational data as input. This, as argued by the paper, is an important bottleneck in models that aim to forecast stochastic dynamical systems in natural science, including biology or material sciences. In such cases, data collection is often destructive, making the collection of paired time-series data impossible. The paper suggests that existing methods including conditional flow matching or Schrodinger Bridges are limited in this regime, and mini-batch optimal transport also struggles when applied to this sparse, unpaired data regime. These limitations include variance explotion, mismatched pairings and training instability.

To tackle these instabilities, the authors introduce Conditional Variable Flow Matching, which leverages a flow definition that operates both over state and conditioning variables. CVFM introduces two stabilization mechanisms: First, an anisotropic conditional Wass. distance that penalizes transports across the conditioning dimension; and, second, a conditioning mismatch kernel that dynamically downweights high variance and mismatched condition pairings during minibatch OT.

The paper is rigorously justified (albeit I am not an expert in many of these topics so I could have missed something), and it proves that OT over the conditioning variable is needed for consistent fields. Empirically, the method is evaluated on 2D synthetic datasets, on higher-dimensional MNIST2fashion domain transfer, and a more complex material science case study. The supplementary materials are of really high value, including extensive proofs, visualizations, code, and extra validation. This supplementary greatly enhances reproducibility and the pedagogical value of the paper as a whole.

**Audience:**

Yes

**Audience Explanation:**

The topics tackled in this paper are very relevant to the TMLR community and beyond, particularly those interested in AI4science, generative modelling, time series, OT, and flow matching.

Extending simulation-free generative models to handle continuous, unpaired conditional distributions is a known bottleneck in fields where data collection is destructive, such as materials science. The authors clearly identify why existing mini-batch optimal transport methods fail in this sparse-data regime (variance explosion) and offer a  grounded, practical solution via the conditioning mismatch kernel. The inclusion of an exceptional supplementary package (source code, a guided tutorial, and visualizations of the dynamic optimal transport) makes this paper not only a solid methodological contribution but also an educational resource. Even if some of the empirical claims require changes, the findings regarding variance reduction in conditional flow matching will be of interest to the TMLR community.

**Broader Impact Concerns:**

There are no apparent negative ethical, societal, or dual-use implications that would require a dedicated Broader Impact Statement. The advancement of robust methods for modeling physical systems is generally beneficial and aligns with the goals of AI4Science.

**Claims And Evidence:**

No

**Claims Explanation:**

The paper procides rigorous proofs and compelling evidence regarding the variance reduction capabilities of the kernel, but some claims are not fully supported by evidence, at least in my understanding:

1. The paper claims that the proposed method outperforms baselines in modelling the dynamics of spinoidal decompositions. However, this is primarily compared against simple non-stochastic baselines like LSTMs. It's difficult to isolate whether the improvements comes from weak baseline definitions or due to the benefits of CVSFM as a method.
2. The MNIST2Fashion dataset is insufficient to proof that the method works on high-dimensional continuous conditioning, as these datasets have discrete conditions. It remains to be validated that high dimensional conditioning really works on the CVFM regime.
3. The paper claims that CVFM achieves improved convergence and more stable training. While these claims are relatively well validated, the entire model requires solving mini-batch OT, which can be really expensive. It'd be very valuable to understand the computational costs of the methods, both in time and memory, and how it scales with dataset size and dimensionality.

To fully support the claims, the authors should either provide the missing benchmarks and continuous high-dimensional evidence, or appropriately narrow the scope of their claims in the text to match the provided evidence.

**Requested Changes:**

- The paper is a bit hard to follow and some rewrite would be welcome (particularly on sections 2 and 3) to make it more accessible to the general reader. It lacks clarity sometimes.
- The claims around the alleged improvements wrt the baselines in the material science experiments should be toned down, or more validation should be provided, with appropriate baselines that are not destined to fail.
- The high-dimensional continuous conditioning claims need to be either toned down significantly or validated empirically with high-dimensional continuous conditioning problems. The experiment of MNIST2Fashion, while compelling, fails to validate this adequately.
- The paper requires computational benchmarks to validate the claims of convergence appropriately, and to understand cost (time and memory) and scalability with respect to dataset size and dimensionality.

---

> ### Author Response · Authors · 2026-03-21
>
> Thanks for your careful review and useful feedback! We've incorporated your suggestions to strengthen the paper and clarify our claims.
>
> 1. We have added several changes to improve the paper's readability. First, we have updated the figure and table captions to be self contained (e.g., to fully define variables and quantities used within the figures and motivate their contents). For example, in Figure 2 we define the value $\eta$. Additionally, we have expanded the motivation provided in the introduction and added two new sections ("Related Works" and "Limitations") to clarify the positioning of this work in the literature as well as to clarify directions of future improvement. We also added Section F (Hyperparameter Guidance) which consolidates the results from the various ablation studies into a singular location and, hopefully, provides guidance for selecting hyperparameters to a potential user.
>
> 2. We appreciate the critique. We would like to note that in the materials dynamics example we also benchmark against Neural ODEs and alternate flow matching based methods (such as Kerrigan et al's T-COT-FM as well as a Score and Flow Matching extension to their method which we implemented). Although we agree this isn't an exhaustive list of available methods for conditional generation, it does reflect state-of-the-art strategies for both the setting with unpaired conditional samples and standard dynamical systems settings. Neural ODEs in particular are a commonly utilized strategy for modeling continuous-time dynamical systems, typically exhibiting much better stability over long, irregular rollouts than autoregressive models like LSTMs.
>
>    In our comparison (Fig. E.10 and Table 2), the Neural ODE (same model backbone as CVFM) and LSTM baselines were given privileged access to the complete, paired trajectory data during training (not supplied to CVFM). All models were evaluated on their ability to predict the entire trajectory rolling out from $t=0$. Even against these privileged baselines, our CVSFM variant achieves lower mean absolute error across these long rollouts. We attribute this specifically to the SDE formulation: the Cahn-Hilliard equation exhibits stiff dynamics where small approximation errors easily compound, pushing deterministic models off the data manifold in early time points. In contrast, CVSFM incorporates a learned score function acting as a restorative vector field, continuously correcting numerical drift by pulling the trajectory back toward high-density regions.
>
> 	However, we agree that the performance improvements, while exciting, are marginal. We revised the materials dynamics discussion  in Sec. 5 to clarify exactly how the error is calculated across long rollouts, explicitly detail the restorative benefits of the score function, and tone down our performance claims to appropriately respect the strength of these baselines.
>
> 3. We appreciate the desire for rigorous validation here, and you are entirely correct. The MNIST-FashionMNIST experiment only validates CVFM in the high-dimensional discrete conditioning setting. We have taken your advice and significantly toned down our claims regarding this point in the manuscript. We revised the text to explicitly state that our continuous conditioning claims are currently validated in low-dimensional spaces (such as the 5D PCA space of the materials study), and we explicitly listed scaling to high-dimensional continuous transport as an open challenge in our new Limitations section.
>
> 4. We added App. G to the paper discussing computational cost and scalability of CVFM. We compare an implementation using just the conditioning mismatch kernel (which we observe can demonstrate competitive performance even in isolation, e.g. Fig. E.5) with CVFM implemented with exact OT and entropically regularized (Sinkhorn) OT. As shown in the newly added Fig. G.1, the exact OT cost scales super-cubically ($\mathcal{O}(B^3 \log B)$), severely impacting runtime for large batch sizes (we predictably observe nearly an order of magnitude increase in cost from $B=64$ to $B=512$). This cost can be reduced by using the Sinkhorn algorithm and nearly completely eliminated by just using the kernel. As shown in the second row of Fig. G.1 (and in Fig. E.5), using these in-exact OT strategies produces comparable performance, especially when the batch size is high.
>
> 	We additionally added text to App. G clarifying that the OT overhead scales strictly linearly ($\mathcal{O}(D)$) with dimensionality. Finally, we note that peak GPU memory is bottlenecked by the network itself rather than the OT solve, as the pairwise distance matrix is negligible for standard batch sizes.

---

### Review · Reviewer_FWhu · 2026-02-25

**Summary Of Contributions:**

This paper studies an important and practically relevant problem: learning stochastic conditional dynamics from sparse, unpaired observations with continuous conditioning variables. The proposed method, Conditional Variable Flow Matching (CVFM), extends flow matching to settings where the conditioning variable is not aligned across mini-batches, and introduces a conditioning-mismatch kernel together with a conditional penalty in the transport cost to stabilize training. The paper’s central motivation is that standard mini-batch couplings in conditional settings can induce severe variance and mismatch, especially under sparse and unpaired scientific data collection regimes.

A notable contribution is the paper’s attempt to connect theory, algorithm design, and scientific applications. The authors provide theoretical statements motivating why local consistency-identity coupling in the conditioning dimension matters for amortized conditional vector field learning, and they derive an error bound (Theorem 3.2) that links gradient mismatch to conditioning mismatch. The empirical section spans toy benchmarks, a scientific stochastic dynamics task (spinodal decomposition microstructure evolution), and an additional higher-dimensional conditional image-domain experiment in the appendix.

Key strengths:
- The research problem setting is quite important and interesting. Sparse, unpaired, condition-dependent observations are common in scientific settings.
- The method proposed is not merely “conditional FM + engineering tricks” but directly targets the conditional mismatch in mini-batch transport.
- The paper provides both theoretical motivation and multiple empirical comparisons.
- The materials microstructure dynamics example makes the contribution more compelling than a toy-only paper.

Key weaknesses:
- While the theory is helpful, the practical algorithm relies on approximations (like those mentioned in mini-batch OT and kernel weighting), and the exact boundary conditions under which the method remains stable are not fully characterized.
- Because stability and sensitivity are central claims, more systematic ablations over kernel and OT-related hyperparameters would strengthen confidence.
- For the materials case, PCA-space trajectory and distribution metrics are reasonable, but more direct physics-aware metrics would strengthen the scientific claims.
- Novelty boundary vs. concurrent conditional OT or FM methods could be clarified further, especially in terms of what gains come from the kernelized mismatch correction versus other implementation choices.

**Additional Comments:**

This is a strong and timely paper in terms of problem choice and methodological relevance. The most compelling part of the submission is not merely the introduction of a new conditional FM variant, but rather the identification and treatment of conditioning mismatch as a practical source of instability in sparse unpaired conditional dynamics learning.

**Audience:**

Yes

**Audience Explanation:**

Yes, definitely. I expect meaningful interest from multiple communities, including:
- Generative modeling, flow matching, OT-based learning
- Scientific machine learning (SciML)
- Stochastic process modeling and surrogate modeling for scientific systems
- Researchers working on learning from sparse, unpaired, or partially observed trajectories

Importantly, the contribution is not just “another flow matching variant.” It addresses a real failure mode in conditional mini-batch transport under unpaired observations, which is a recurring issue in practical scientific data modeling. The extension to stochastic dynamics further broadens the interest of researchers modeling noisy physical processes.

**Broader Impact Concerns:**

I do not see major direct ethical concerns or obvious dual-use risks based on the work as described. However, the paper does not include a limitations or broader impact discussion, I would recommend adding a brief paragraph addressing them.

**Claims And Evidence:**

Yes

**Claims Explanation:**

The paper’s central claims appear to be:
1. Conditional mismatch in mini-batch couplings is a major source of variance and instability in conditional flow matching under sparse unpaired data.
2. CVFM mitigates this issue by kernel-weighted conditional mismatch correction and anisotropic transport costs.
3. This improves optimization stability and empirical performance, including in a scientific stochastic dynamics benchmark.

Most of the core claims are supported by reasonably convincing and fairly clear evidence, though some claims would benefit from stronger supporting analyses.

I would characterize the evidence as substantially supportive for the main methodological claims, but not yet exhaustive, especially regarding robustness, sensitivity, and broader applicability.

These claims are supported in several ways:

### Why the evidence is convincing in the main line
- Theoretical support is provided, rather than purely heuristic motivation. Theorem 3.2 (and associated derivations) explicitly links mismatch-related quantities to training objective error, which strengthens the method’s conceptual grounding.
- Variance-focused diagnostics are included (e.g., objective target variance comparisons in the paper), which directly address the claimed mechanism of improvement (reduced instability), rather than only reporting final performance.
- The empirical evidence is multi-scenario, including low-dimensional controlled cases and a more realistic scientific dynamics example. This is stronger than relying solely on toy settings.

### Why the evidence is not fully complete
- Robustness boundaries are not fully mapped. Since the method’s main selling point is stability, a more comprehensive stress test over hyperparameters (like kernel bandwidth, transport penalty coefficient, batch size, etc.) would make the evidence more complete.
- The method appears especially targeted to settings where the conditioning marginal behaves in a specific way. The practical implications of violating these assumptions are not fully explored.
- Scientific validity could be more directly measured in the materials setting with domain-specific structure metrics beyond latent or proxy summaries.

So, in summary, the paper does provide accurate and relevant evidence for its main claims, but there is still room to improve the depth and breadth of validation.

**Requested Changes:**

Below, I separate suggested revisions into (A) more critical or likely to affect acceptance confidence and (B) strengthening revisions.

### A. Higher-priority adjustments
1. Add a more systematic robustness/sensitivity analysis (kernel bandwidth, mismatch penalty coefficient, batch size, OT solver choice, solver regularization)
   - Priority: Potentially critical
   - Rationale: stability is a core claim in the paper. Robustness evidence should match the centrality of that claim.

2. Clarify novelty and empirical differences vs. closely related or concurrent methods (e.g., conditional OT/FM variants)
   - Priority: Potentially critical
   - Rationale: This directly affects how the contribution is assessed by reviewers.

3. Include failure modes or degradation regimes
   - Examples: highly sparse conditioning support, high-dimensional conditioning, strong condition shift, non-stationary conditioning marginals
   - Priority: Potentially critical

### B. Strengthening revisions
4. Add more domain-specific evaluation metrics in the materials science case
5. Report computational cost and efficiency trade-offs
6. Promote key ablations from the appendix to the main text

---

> ### Author Response · Authors · 2026-03-21
>
> Thank you for your comprehensive feedback. We have revised the paper to directly address the points you raised (higher-priority and strengthening revisions):
>
> Systematic Sensitivity Analysis:
>
> - We agree that a systematic treatment of robustness is crucial. To complement existing ablations on batch size (Sec. 3.3, Fig. 2) and our kernel/penalty sweeps (App. E, Figs. E.2, E.4 - E.6), we have added:
>
> - OT Solver: In Table 1, we now systematically report results for exact OT vs. entropically-regularized (Sinkhorn) solvers. We added a computational cost analysis (App. G) comparing scalability and runtime trade-offs of these solvers against the mismatch kernel in isolation. As shown in the new Fig. G.1, OT solve cost scales poorly with batch size (nearly an order of magnitude increase from $B=64$ to $B=512$). However, this cost can be minimized by Sinkhorn and nearly eliminated by using the kernel in isolation. As shown in Fig. G.1 and Fig. E.5, using Sinkhorn or the kernel alone results in competitive performance with superior scaling. We also included Wasserstein Flow Matching (Haviv et al., 2024) as a benchmark for discrete conditioning (similar to CGFM).
>
> - Hyperparameter Guidance: To increase accessibility, we synthesized our analyses into a Hyperparameter Guidance section (App. F), providing practical advice on setting $\sigma_y$ and $\eta$.
>
> Framing and Related Work:
>
> - Excellent point. We added a dedicated section (Sec. 4) discussing concurrent and related methods. This contextualizes CVFM within the broader family of conditional transport methods and those directly applicable to stochastic dynamics with sparse unpaired conditioned samples.
>
> Limitations:
>
> - We added a dedicated Limitations section (Sec. 5.1) immediately prior to the conclusion. In this new section, we explicitly outline the boundary conditions and challenges you mentioned. Specifically, we note that the Euclidean distance metric utilized by our kernel may degrade in high-dimensional conditioning settings or under severe condition shifts. We also explicitly note that our theoretical derivation (and resulting stability of the method) relies on the assumption of stationary conditioning marginals across time ($q(y_0) \approx q(y_1)$), and that extending the framework to time-varying condition marginals remains an important open area of research.
>
> Evaluation Metrics:
>
> - We appreciate this feedback. To clarify, the predicted state variables (PC scores) are derived from 2-point spatial correlations of Cahn-Hilliard simulations. In materials informatics, these 2-point statistics act as rigorous, domain-specific features that inherently capture fundamental physical morphological features (e.g., volume fraction and relative spatial positioning of constituents) while accounting for periodic symmetries. These features are popular in the community because they represent the complexity of the system while enabling the adoption of standard metrics and losses.
>
> - We agree the evaluation of dynamic distributions could be strengthened. We augmented Table 1 with Maximum Mean Discrepancy (MMD) and Energy Distance (ED) for a more comprehensive statistical evaluation. Methodology discussion for these metrics is now in App. D.3.
>
> Computational Cost:
>
> - We added a dedicated cost and scalability section (App. G), including Figure G.1, which evaluates wall-clock training time against model performance across 2D benchmarks. This highlights efficiency trade-offs: exact OT exhibits poor scalability at large batch sizes, while Sinkhorn provides a faster but still overhead-heavy alternative. Crucially, bypassing the OT solver entirely via our mismatch kernel provides a highly scalable, lightweight alternative achieving competitive $W_2$ performance at a fraction of the cost.
>
> Ablations:
>
> - We certainly agree there are many ablations within the paper. Although, to strike a balance between narrative flow, page limits, and empirical rigor, we have ensured that the most critical ablations (Objective Target Variance (OTV)) is prominently featured in the main text (Sec. 3.3, Fig. 2 and Table 3), explicitly demonstrating the method's stability in sparse regimes.
>
> - We have promoted our explicit Algorithm blocks (previously Alg. 1 (CVFM) and 2 (CVSFM) in Appendix) to the main text. While we kept the more granular hyperparameter sweeps ($\sigma_y$ and $\eta$) and kernel-choice ablations in the Appendix, we have added stronger phrasing in the main text directing interested readers as to their locations.
>
> Broader Impacts:
>
> - Lastly, regarding broader impact, we appreciate the reviewer noting the absence of obvious dual-use or ethical risks. Because this work focuses on foundational methodological improvements, societal impacts are highly speculative and dependent on downstream implementations. To best utilize space, we opted to expand on technical Limitations (Sec. 5.1) rather than adding a speculative Broader Impacts paragraph, aligning with the principal goals of the paper.

---

### Review · Reviewer_nP5t · 2026-03-10

**Summary Of Contributions:**

The authors introduce Conditional Variable Flow Matching (CVFM), a novel method for learning transport dynamics amortized across the space of conditional densities. Notably, their proposed approach assumes no pairing across conditional marginal distributions, addressing a limitation of existing flow-based generative models when working with sparse and continuous conditions. To address this, the authors propose the use of a conditioning mismatch kernel alongside a distributional loss (Wasserstein loss) to reweight the conditional optimal transport objective. Together with the flow matching objective, the authors demonstrate that CVFM exhibits stable training with fast convergence. Through empirical experiments, the authors demonstrate that CVFM yields competitive performance on toy examples, image-to-image translation, and on simulated material dynamics datasets. Overall, the authors present a compelling method for modeling conditional transport dynamics across _unpaired_ conditional densities.

Strengths:
- Authors provide a strong theoretical backing and support for their proposed method.
- Thorough set of synthetic empirical experiments ranging from 2D toy examples, to simulated material dynamics simulations, and images, with many visuals, tables, and figures to support their claims.

Weaknesses:
- At times, paper is hard to follow and presentation of main findings could benefit from better organization; e.g. experiments section is unclear at times, table/figure captions in experiment section could be more self-contained, further explanation of baseline and experiment implementations in main text would be useful, etc.
- Overall motivation for learning conditional transport dynamics for unpaired conditional densities feels a little weak, but I think this can be clarified and improved (see additional comments and suggested related work for examples).
Some relevant literature, background, and baseline(s) are missing (see below).

**Additional Comments:**

Questions:
- In section 3.1, the authors state: “we further assume that the conditional joint probability path decomposes as $p_t(x, y | z, w) = p_t(x | z)p_t(y | w)$ … “. However, the authors also state that they “assume that the joint distribution at intermediate times, $p_t(x, y)$, retains a dependency between $x$ and $y$.” In the previous statement, it seems the authors implicitly assume _independence_ between $x$ and $y$, while the modeling goal defined in equations 6 assumes _dependence_, $p(x | y)$ and the latter statement also implies _dependence_. Could the authors clarify this, as it appears as a discrepancy in modeling assumptions?
- What is the final loss function used for training the model (vector field)? In figure 1, the authors state that “CVFM integrates a conditioning mismatch kernel with a conditional wasserstein distance in the training objective”, but this final loss is not clearly stated in the methods.

Related Works:
- The authors of Meta Flow Matching (MFM) [1] introduce the general form of “conditional generative flow matching” (CGFM) in [1, section 2.2]. In fact, I think this may be very close to what you label as CGFM in this work; i.e. _ideal_ paired conditional distributions. Moreover, [1] also introduces an approach which amortizes flow matching on the space of densities and is related to CVFM, however, assuming paired conditional densities. [1] should be cited in general and included in a related work section. MFM should also likely be considered as a baseline, but I would prioritize the method which I outline in the third bullet point (see below) as it is directly applicable to the unpaired conditional distribution setting.
- [4] extends a CGFM-type framework to the multi-time point (multi-marginal) OT setting, and should be cited and mentioned in related work.
- The authors in [2] introduce Wasserstein Flow Matching (WFM), a general framework for lifting flow matching to the space of distributions. Notably, this method can be applied to the setting of unpaired conditional distributions [2, sec 3.2] as considered by you in this work. I suggest the authors implement WFM on general distributions as a baseline as well as include a discussion of this method in related work.
- In a similar manner, [3] introduce an analogous approach for learning conditional transport maps across unpaired distributions. I note this is concurrent work and does not need to be included, but I mention it here for interest and given the related context.

Minor comments:
- Should Equation 6 have two integrals, since you are integrating over both $z$ and $w$)?


**References**:

[1] Atanackovic, Lazar, et al. "Meta flow matching: Integrating vector fields on the wasserstein manifold." 2024

[2] Haviv, Doron, et al. "Wasserstein flow matching: Generative modeling over families of distributions." 2024

[3] Fishman, Nic, et al. "Distribution-Conditioned Transport." 2026

[4] Rohbeck, Martin, et al. "Modeling complex system dynamics with flow matching across time and conditions." 2025

**Audience:**

Yes

**Audience Explanation:**

Yes. The problem of modeling conditional dynamics has been a longstanding problem in many scientific applications. More generally, recent work has formalized the problem considered in this work more generally as conditional generative modeling on the space of distributions [1, 2, 3] and has received attention in biological applications. As a result, I believe individuals of the TMLR audience will be interested in this paper.

**Broader Impact Concerns:**

No concerns. This paper presents primarily a theoretical approach for conditional generative modeling. Hence does not present any immediate broader impact concerns.

**Claims And Evidence:**

Yes

**Claims Explanation:**

In general **yes**, the claims are supported. I provide some suggestions (listed in requested changes and additional comments) to further strengthen the claims or where relaxing claims is encouraged. Below I provide a summary of the main claims of this work and outline their respective _support_.

Claims:
1. CVFM exhibits better performance on toy problems compared to counterpart baselines
2. CVFM exhibits Improved convergence compared to counterpart baselines.
3. CVFM is better than baselines for modeling material dynamics / generating dynamics of microstructure evolution in high dimensions.
4. CVFM is better than baselines on image-to-image mapping
5. CVFM can approximate SB/stochastic dynamics.

Support:
1. **Mostly Supported**. The authors designed toy systems/problems specifically tailored to test their claims; i.e. that their method, CVFM, is better suited for modeling flows across unpaired conditional distributions. They demonstrated empirically that CVFM yields lower distributional error (2-Wasserstein distance) than counterpart baselines, when conditional distributional pairings are not provided to the model. However, the authors only consider simple baselines and should consider Wasserstein Flow Matching over general distributions [2, sec 3.2] as a possible counterpart and competitive method to compare with.
2. **Mostly supported**. The authors claim that CVFM exhibits improved convergence, and conduct experiments to empirically demonstrate convergence compared to counterpart baselines. I suggest the authors be clear in text that improved convergence is specific to the setting they consider in this work; i.e. unpaired conditional distributions.
3. **Mostly supported**. The authors design a “synthetic” experiment for material dynamics using simulated data and empirically demonstrate that CVFM yields favourable performance compared to counterpart baselines. However, the authors do not use “real” data, on simulation generated data. As a result, I suggest the authors state this limitation more transparently in the introduction and respective experiment section. Furthermore, results are reported in principle component (PC) space (Table 2). It is unclear whether CVFM was trained on the high dimensional space or PC space.
4. **Weakly supported**. CVFM’s ability to improve performance for image-to-image transformation is only weakly supported in the appendix. Only one baseline is considered and results are only shown in the appendix. I suggest the authors either relax this claim, which is stated in the introduction, or include a more comprehensive empirical study of this task and include results in the main text.
5. **Mostly supported**. The authors demonstrate empirically that their method, CVFM, is amenable to modeling conditional Schrodinger bridges. However, these results lack a bit of clarity on how CVFM was trained in the stochastic setting, making it hard to assess the strength of the claim.

To add, the authors also provide thorough theoretical support and backing for their proposed method, strengthening the claims/contributions of their proposed method.

**Requested Changes:**

Experiments:
- Wasserstein flow matching (WFM) (specifically, WFM over general distributions [2, sec 3.2]) should be added as a baseline and should be applicable to most of the experiments considered in this work.
- Only one metric is considered. I suggest that the authors add an additional metric, such as Maximum Mean Discrepancy (MMD), or energy distance.

In text changes and clarifications:
- The authors should include an algorithm box in the main text showing different versions of CVFM (ODE and SDE versions).
- Clarification of some assumption in section 3.1 (see questions in additional comments)
- The authors should include an explicit related works section.
- Moreover, the author should add some highly related works on conditional generative flow matching for dynamics (see additional comments below).
- There is no discussion of limitations and future work in the main text. I suggest the authors introduce a thorough discussion of limitations, and add discussion of future research directions.
- In text clarification regarding improved convergence; i.e. specific to the setting considered in this work (unpaired conditional distributions).
- Clarification (in the introduction) that the material dynamics experiments are conducted using simulated data and clarification on high-dimensional versus low dimensions PC evaluations.
- Clarification of the final loss used to train CVFM (see question 2 below).

---

> ### Author Response · Authors · 2026-03-21
>
> We thank the reviewer for the comprehensive review of our work. We agree with all major points raised and have adjusted the paper accordingly.
>
> Experiments:
>
> - Thank you for pointing us to Haviv et al. (2024). We added Wasserstein Flow Matching (WFM) as an additional baseline in Table 1 for all applicable discrete conditioning cases. We observe WFM yields mixed, competitive performance on these discrete tasks (trailing CVFM Gaussian-Moons, narrowly outperforming Gaussian-Gaussian). Ultimately, both are outperformed by CGFM, which utilizes direct, microscopic conditional knowledge during training.
>
> - WFM's measure-wise formulation is structurally incompatible with our continuous problem settings. WFM flows between empirical distributions by treating them as macroscopic point clouds. In our continuous settings, each observation $x$ carries a unique continuous conditioning variable $y$, meaning the empirical sample size for any specific condition $y$ is exactly $N=1$ (while the true distribution isn't a dirac, our sparse datasets only contain one $x$ per $y$). This precludes WFM from computing meaningful self-attention scores across the distribution. Arbitrarily binning the continuous conditioning variables to create valid point clouds ($N>1$) would fundamentally alter the problem and destroy the continuous nature of the physical parameter space. We have detailed this architectural distinction in our new Related Works (Sec. 4).
>
> - We included both MMD and ED as additional metrics in Table 1 for relevant case studies and expanded our experimental methodology discussion in App. D.3. In the process of adding additional metrics, we re-ran the complete experiment in Table 1 and updated it with these recent results. They consistently align with the $W_2$ evaluations, further highlighting that CVFM's inherent variance reduction improves conditional generation across all mappings.
>
> Clarifications:
>
> - We moved our explicit algorithm blocks from the Appendix to the end of Sec. 3.3, displaying a unified algorithm covering both ODE/SDE variants directly in the main body.
>
> - We made the assumptions in Sec. 3.1 more explicit in the introductory paragraphs. Specifically, we assume independence in conditional trajectory paths upon construction (e.g., $p(y|w)$ and $p(x|z)$), although dependence is obtained by the expectation with respect to the joint empirical distribution $q(z,w)$.
>
> - We added an explicit Related Works section (Sec. 4) augmenting the embedded discussion previously in the introduction. This explicitly contextualizes CVFM alongside Meta Flow Matching, Wasserstein Flow Matching, Distribution-Conditioned Transport, and multi-marginal frameworks to better frame our positioning.
>
> - We added a dedicated Limitations section (Sec. 5.1), highlighting limitations involving additional hyperparameters, time-invariant conditioning assumption, and Euclidean distance metrics potentially degrading in higher dimensions.
>
> - We added a statement to Sec. 5 (improved convergence subsection) noting improvements are relative to other methods applicable to the continuous unpaired conditional setting (i.e., COT-FM / T-COT-FM and variants).
>
> - We reframed the introduction's final paragraph to highlight earlier that the materials dynamics involve phase-field simulated microstructure evolution pathways conditioned on manufacturing parameters. In Sec. 5, we now specify that 2-point spatial correlations were compressed to a 5-dim representation via PCA to define state variables, and all error metrics were evaluated in PC space. These 2-point statistics act as rigorous, domain-specific features inherently capturing fundamental physical morphological features (e.g., volume fraction, relative spatial positioning) while accounting for periodic symmetries.
>
> - We agree the claims regarding the image domain translation task were weakly supported in the main text. Following your suggestion, we removed the domain translation performance claims from the Introduction and main body, allowing the experiment to stand purely as a supplementary interrogation of competing methods in App. E.3.
>
> - The final CVFM training loss is now prominently highlighted in the new Alg. 1, in addition to Eq. (7). Algorithm 1's OT solves leverage the anisotropic cost in Eq. (9).
>
> - Corrected Eq. (6) to include a double integral for clarity.
>
> - We agree the motivation for learning dynamics across unpaired, continuous densities was underdeveloped regarding broader conditional generative modeling. We revised the Introduction and Related Works (Sec. 4) to explicitly bridge our real-world motivation (e.g., destructive testing) with formalisms in recent works like WFM and Distribution-Conditioned Transport. By establishing how concurrent methods formalize transport of distribution families, we more sharply motivate CVFM: it is the necessary step to prevent variance explosion when measure-wise transport formalisms are forced into the limit of continuous conditional parameters.

---

### Decision · Action_Editor_RYEr · 2026-04-14

**Recommendation:** Accept as is

**Audience:**

Yes

**Audience Explanation:**

Unpaired translation is an open problem in the machine learning community.
Most methods are linked with generative AI which is of interest for the global community.

**Claims And Evidence:**

Yes

**Claims Explanation:**

In this paper, the authors introduce Conditional Variable Flow Matching (CVFM), a novel method for learning transport dynamics amortized across the space of conditional densities. The authors evaluate their methods on image-to-image translation, and on simulated material dynamics datasets. The text now reflects accurately the contribution after discussion with the reviewers. As Reviewer mW6b puts it "They have added additional experiments/baselines to better support their empirical claims, added discussion to relevant related works, amended text that included overstated claims".